# A non-canonical role of somatic Cyclin D/CYD-1 in oogenesis and in maintenance of reproductive fidelity, dependent on the FOXO/DAF-16 activation state

Umanshi Rautela[1☯¤]*, Gautam Chandra Sarkar[1☯¤], Ayushi Chaudhary[1], Debalina Chatterjee[1], Mohtashim Rosh[2], Aneeshkumar G. Arimbasseri[2], Arnab Mukhopadhyay[1]*

1 Molecular Aging Laboratory, National Institute of Immunology, Aruna Asaf Ali Marg, New Delhi, India,
2 Molecular Genetics Laboratory, National Institute of Immunology, Aruna Asaf Ali Marg, New Delhi, India

☯ These authors contributed equally to this work.
¤ Current address: Department of Pediatrics, Washington University School of Medicine, St. Louis, Missouri, United States of America
* umanshi@nii.ac.in (UR); arnab@nii.ac.in (AM)

**Data Availability Statement:** The data reported in this manuscript is available in S1 Table. The S2 Table and S3 Table have the values for the RNA-

## Abstract

For the optimal survival of a species, an organism coordinates its reproductive decisions with the nutrient availability of its niche. Thus, nutrient-sensing pathways like insulin-IGF-1 signaling (IIS) play an important role in modulating cell division, oogenesis, and reproductive aging. Lowering of the IIS leads to the activation of the downstream FOXO transcription factor (TF) DAF-16 in *Caenorhabditis elegans* which promotes oocyte quality and delays reproductive aging. However, less is known about how the IIS axis responds to changes in cell cycle proteins, particularly in the somatic tissues. Here, we show a new aspect of the regulation of the germline by this nutrient-sensing axis. First, we show that the canonical G1-S cyclin, Cyclin D/CYD-1, regulates reproductive fidelity from the uterine tissue of wild-type worms. Then, we show that knocking down *cyd-1* in the uterine tissue of an IIS receptor mutant arrests oogenesis at the pachytene stage of meiosis-1 in a DAF-16-dependent manner. We observe activated DAF-16-dependent deterioration of the somatic gonadal tissues like the sheath cells, and transcriptional de-regulation of the sperm-to-oocyte switch genes which may be the underlying reason for the absence of oogenesis. Deleting DAF-16 releases the arrest and leads to restoration of the somatic gonad but poor-quality oocytes are produced. Together, our study reveals the unrecognized cell non-autonomous interaction of Cyclin D/CYD-1 and FOXO/DAF-16 in the regulation of oogenesis and reproductive fidelity.

## Author summary

The conserved insulin-IGF-1 signaling (IIS) pathway in *Caenorhabditis elegans* is a neuro-endocrine cascade that surveils the availability of nutrients as well as exposure to

seq data. The RNA seq reads have been submitted https://www.ebi.ac.uk/biostudies/arrayexpress with the accession number E-MTAB-13172.

**Funding:** This project was partly funded by the National Bioscience Award for Career Development (BT/HRD/NBA/38/04/2016) and extramural grant (BT/PR27603/GET/119/267/2018) from the Department of Biotechnology, Government of India (https://dbtindia.gov.in/), Science and Engineering Research Board-Science and Technology Award for Research (SERB-STAR) award (STR/2019/000064), Jagadish Chandra Bose National Fellowship (JCB/2022/000021) and extramural grant (CRG/2022/000525) from the Ministry of Science and Technology, Government of India (https://serb.gov.in/page), as well as core funding from the National Institute of Immunology (to AM). GCS is supported by an ICMR SRF fellowship (RMBH/FW/2020/19), and UR by DBT-JRF fellowship DBT/2018/NII/1035. The funders had no role in study design, data collection and analysis, decision to publish, or preparation of the manuscript.

**Competing interests:** The authors have declared that no competing interests exist.

environmental and cellular stressors to regulate somatic development, reproduction and aging, both cell autonomously and non-autonomously. Lowering IIS flux activates the downstream FOXO transcription factor DAF-16 that mediates most of these responses. How this pathway responds to perturbations in cell cycle proteins to influence reproductive decision is less known. Here we show that the G1-S phase cyclin, Cyclin D/CYD-1 and the IIS nutrient-sensing axis crosstalk to regulate oogenesis and germ cell quality. In the wild-type worms, we found that CYD-1 in the uterine tissue regulates oocyte health and reproductive fidelity. Knocking down *cyd-1*, only in the somatic uterine tissue of IIS mutants, leads to a DAF-16-dependent arrest of oogenesis. Interestingly, we observed DAF-16-dependent destruction of the somatic gonadal tissues and de-regulation of the sperm-to-oocyte switch genes when CYD-1 is depleted in the uterus of animals with lower IIS. We speculate that defects in cell cycle proteins in the uterus may be interpreted by activated DAF-16 as a potential threat to progeny health that is unlikely to be rectified. So, our study unveils a new function of activated FOXO/DAF-16 and Cyclin D/CYD-1 in maintaining oocyte health and reproductive fidelity.

## Introduction

Germline is the most precious tissue of an organism that ensures perpetuation of a species. However, with aberrant metabolism brought about by aging and metabolic disorders, the reproductive tissues deteriorate leading to poor quality of oocytes, adversely affecting reproductive outcomes and progeny fitness [1–5]. To ensure optimal growth of the gametes and the fertilized oocytes, the process of oogenesis takes inputs from other reproductive and somatic niches in the body. For example, oogenesis is firmly regulated by the nutrient-sensing [6–8] and stress-responsive pathways [9,10]. In *C. elegans*, the insulin/IGF-1 signaling (IIS) pathway, a conserved neuro-endocrine signaling axis that is central to nutrient sensing and stress resilience [11–14], regulates both somatic [15,16] as well as reproductive aging [17,18] in a cell-autonomous as well as non-autonomous manner [18–22]. In mammals, IIS is crucial for the activation, and growth of the primordial oocyte follicles [23–25]. The imbalance in IIS underpins pathophysiologies of multiple ovarian dysfunctions including PCOS and infertility [26–29]. The IIS-PI3K-AKT axis, when activated, maintains the FOXO transcription factor in the cytoplasm through inhibitory phosphorylation [30–32]. The overactivation of the PI3K-AKT pathway that is downstream of the IIS receptor has been shown to cause global activation of the oocyte follicles and depletes the ovarian reserve, leading to premature reproductive aging and infertility in mice [33]. On the other hand, when the signaling through the IIS pathway is low, FOXO is released from its cytoplasmic anchor and enters the nucleus to activate gene expression [14,30–32]. It has been noticed that FOXO activation during the early stages of oocyte development preserves the ovarian reserve and extends reproductive capacity in mice [34,35]. In *C. elegans*, where the mechanisms of aging are relatively well worked out [36–38] mutations in the *daf-2* (the IIS receptor ortholog) lower signaling flux through this pathway, leading to a long life and a slower reproductive aging [17,18,39]. The *daf-2* worms have a delayed decline in oocyte quality [18,39] and germline stem cell pool with age [22]. Interestingly, activated DAF-16 is required in the intestine and muscle for oocyte quality maintenance [18] while DAF-16 in the somatic gonad delays age-related germline stem cell loss [22]. It is important to note that in most cases, activated FOXO is pro-longevity and it preserves reproductive fidelity.

To identify unexplored interactors of the IIS pathway, we had earlier performed a reverse genetic screen and identified a cyclin-dependent kinase CDK12/CDK-12 that regulates oogenesis cell non-autonomously [40]. To broaden our understanding of other cell cycle regulators that may have non-canonical roles in reproductive aging and if they crosstalk with the IIS pathway, we screened for cyclins that regulate germline development. Interestingly, this led to the identification of Cyclin D/CYD-1 [a core cell-cycle protein that regulates the G1 to S phase transition] [41,42] as a novel interactor of the IIS pathway. Beyond their involvement in cell cycle progression, cyclins and CDKs exhibit diverse biological functions [43,44]. Cyclin D is particularly noteworthy for its involvement in a broad spectrum of biological functions, encompassing metabolic regulation [45,46], organ development [47,48], DNA damage repair [49], and transcriptional control [50–52].

In this study, we report two novel and unanticipated observations: 1) *cyd-1* knock-down solely in the somatic uterine tissues of wild-type (WT) worms, compromises reproductive fidelity, leading to poor quality oocytes and endomitotic oocytes in the germline, and 2) *cyd-1* knock-down in the uterine tissues causes DAF-16-dependent arrest of the germline in the pachytene stage of meiosis-I specifically in the *daf-2* mutant (where DAF-16 is in an activated state). We also show an unexpected link between activated DAF-16 and dramatic somatic gonad defects with concomitant downregulation of genes important for the sperm-to-oocyte switch that may cause the germline pachytene arrest and halted oogenesis.

Together, our study highlights the non-canonical, cell non-autonomous function of Cyclin D/CYD-1 in reproductive fitness and the unconventional role of activated FOXO/DAF-16 in causing tissue damage to stall oogenesis. Considering the conserved nature of these players, it is tempting to speculate that such mechanisms may be conserved during evolution.

## Results

In a recent study, we described the crosstalk of the cyclin-dependent protein kinase, CDK-12 with the IIS pathway to regulate germ cell development [40]. Interestingly, in that study, the function of the CDK-12 was found to be independent of its canonical cyclin, cyclin K/CCNK-1 [53]. Following up on this study, we asked if other cyclins may crosstalk with the IIS pathway to regulate germline development. For this, we knocked down the annotated cyclins in *C. elegans* in *daf-2(e1370)* (referred to as *daf-2*) and found that the RNAi-mediated depletion/knock-down (KD) of *cyclin D/cyd-1* and *cyclin E/cye-1*, genes that encode core cell-cycle proteins governing the transition from the G1 to S phase, caused a DAF-16 dependent sterility specifically in a *daf-2* mutant (**S1A Fig**). To determine the effect of *cyd-1* and *cye-1* KD on germline development, the gonads of day-1 adults were stained with DAPI to mark the nucleus. The KD of *cye-1* drastically reduced the germ cell number in *daf-2* worms, as is expected from its known function in cell division (**S1B Fig**), whereas *cyd-1* KD demonstrated a non-canonical role in germ cell development without dramatic changes in mitosis, as described below. The loss-of-function *cyd-1* alleles are arrested during early larval development and are sterile [42]. Consequently, they could not be used for our experiments to study interaction with the IIS pathway and so, we focused our efforts on investigating the role of CYD-1 in germ cell development and quality assurance using RNAi.

### CYD-1 maintains oocyte quality and reproductive fidelity in wild-type worms

Since *cyd-1* KD renders *daf-2* worms sterile, we first asked how CYD-1 may influence WT germline where insulin signaling may be considered normal. The *C. elegans* hermaphrodite gonad has two U-shaped tubular arms where the germ line stem cell (GSC) pool resides near

the distal end and divides mitotically. On moving away from the distal end, the germ cells enter meiotic prophase. The sperm formation takes place at L4 stage (sperms are stored in the spermatheca), after which a switch from sperm-to-oocyte fate takes place. Subsequently, germ cells are committed to developing into oocytes [54–56]. The quality of oocytes is ensured by programmed cell death of defective germ cells at the end of pachytene region. A common uterus that carries the fertilized eggs (until they are laid) connects both the gonadal arms [57] (**Fig 1A**; only one gonad arm is shown).

WT worms were grown from larval stage 1 (L1) onwards on control or *cyd-1* RNAi, DAPI stained at day-1 and imaged to observe the germ cell number (**S1C and S1D Fig**). Interestingly, we did not observe changes in the WT mitotic or transition zone germ cell number but found a significant reduction in the pachytene cell numbers (**S1D Fig**). We questioned whether a decrease in the pachytene germ cell number could be due to an increase in germ cell apoptosis. Indeed, we observed an increase in apoptotic germ cell numbers marked as CED-1:: GFP-expressing foci [CED-1 is a transmembrane protein on the surface of sheath cells that engulfs the apoptotic germ cells, thereby marking the dying germ cells [58]] in the gonad of *cyd-1* RNAi worms (**S1E and S1F Fig**). Normally, as the pachytene cells transition to diplotene, most of the defective/damaged germ cells are culled in the turn region of the gonad to ensure that only healthy oocytes mature [59]. The increase in apoptosis upon *cyd-1* KD suggested an increase in damaged germ cells. We therefore evaluated the oocyte quality as a readout of germ cell quality. Firstly, the oocytes were examined morphologically by using differential interference contrast (DIC) microscopy (magnification of 400 X) on day-3 of adulthood. The oocytes were assigned a score (normal, mild or severely defective morphology) based on the presence of cavities, abnormal shape/size, and/or organization (**S1G Fig**). On day-3 of adulthood, WT worms grown on control RNAi showed the proper arrangement of stacked oocytes without any cavities (**Fig 1B and 1C**). However, upon *cyd-1* KD, the oocytes became significantly more misshapen, and the gonads were disorganized, with cavities between oocytes (**Fig 1B and 1C**). Further, we asked if the observed poor oocyte morphology also translates into poor quality of these oocytes. Poor quality oocytes often lead to a reduction in the hatching efficiency of the eggs. In the case of the wild-type worms, the percentage of hatched eggs is known to decrease with age as the oocyte quality declines [59]. Noteworthily, we found a reduction in the percentage of eggs that hatched upon *cyd-1* KD (**Fig 1D**). These observations indicate that CYD-1 is required in wild-type worms to maintain the quality of germ cells and oocytes.

The *C. elegans* self-fertilizing hermaphrodite contains both sperms and oocytes. To assess if the poor oocyte quality upon *cyd-1* KD is dependent on sperm/sperm signals, we mated the *cyd-1* KD hermaphrodites to males grown on control RNAi. Interestingly, both selfed and mated hermaphrodites displayed poor oocyte morphology upon *cyd-1* depletion, implying that the poor oocyte quality is sperm-independent (**Fig 1E and 1F**).

The reproductive span (the number of days the worms can lay live eggs) of the *cyd-1* KD worms was not reduced (in both selfed as well as mated worms) compared to control RNAi-grown worms (**Fig 1G**) although an overall reduction in the brood size was observed (**Fig 1H**). We also observed a greater than 3-fold increase in the incidences of endomitotic oocytes (emos) at day-3 of adulthood in *cyd-1* KD WT worms (**Fig 1I and 1J**). Such emos are often found in the uterus of aged worms, in the post-reproductive phase [60]. We asked if these emos have lost their germ cell fate and acquired somatic fate, similar to ovarian teratomas. To address this, we examined the expression of *unc-119*::GFP, which is expressed pan-neuronally but not in oocytes [61]. Depletion of *cyd-1* resulted in *unc-119p*::GFP expressing emos (**Fig 1K and 1L**), thereby indicating trans-differentiation of these germ cells. We also found the presence of these densely DAPI-stained emos upon *cyd-1* KD in the feminized worms, indicating these are sperm-independent in origin (**S1H Fig**). Interestingly, *cyd-1* KD post late-L4 larval

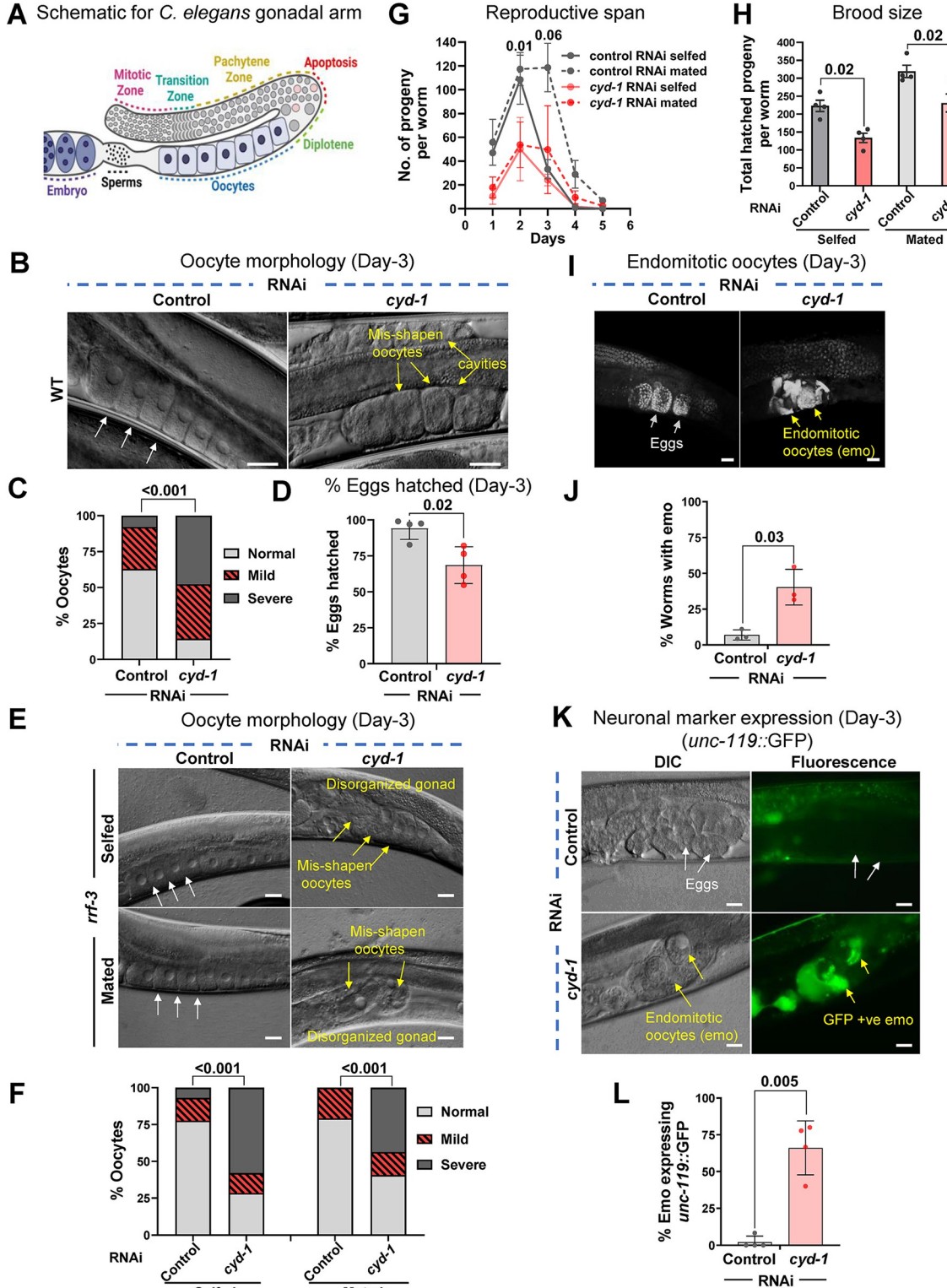

**Fig 1. CYD-1 ensures optimal oocyte quality in wild-type. (A)** A diagrammatic representation of the right arm of *C. elegans* gonad. Schematics in Fig 1A were created using BioRender.com. **(B,C)** DIC images showing oocyte morphology of WT (day-3 adult) grown on control or *cyd-1* RNAi. White arrows mark normal oocytes while yellow arrows mark poor-quality oocytes that have cavities or are misshapen (B). Oocyte quality scores (based on morphology) (C). The quality was categorized as normal, or mild or severe based on its morphology (cavities, shape, organization) as shown in S1G Fig. Combined data from three biological replicates (n ≥ 85) is plotted.

Chi-square analysis was used to compare between groups. **(D)** Percentage of WT eggs (day-3 adult) that hatched on control or *cyd-1* RNAi. Average of four biological replicates (n ≥ 50 for each replicate). Unpaired *t*-test with Welch's correction. **(E,F)** Oocyte morphology in unmated or mated day-3 adult *rrf-3(pk1426)* worms grown on control or *cyd-1* RNAi (E). Oocyte quality scores (based on morphology as shown in S1G Fig) (F). Combined data from three biological repeats (n ≥ 48) is plotted. Chi-square analysis was used to compare between groups. **(G)** The reproductive span of WT (selfed and mated worms) on control or *cyd-1* RNAi. Data from three biological replicates is shown. *P*-value for mated worms on control versus *cyd-1* RNAi at day-2 and day-3 is shown. Unpaired *t*-test with Welch's correction. **(H)** The total number of hatched progenies in WT (selfed and mated worms) grown on control or *cyd-1* RNAi. Average of four biological repeats. Unpaired *t*-test with Welch's correction. **(I,J)** Representative DAPI-stained gonads of WT (day-3 adult) worms grown on control or *cyd-1* RNAi. White arrows mark normal eggs while yellow arrows mark endomitotic oocytes (emos) (I). Quantification of percent endomitotic oocytes (J). Average of three biological replicates (n ≥ 25 for each experiment). Unpaired *t*-test with Welch's correction. **(K,L)** Representative fluorescent images of *unc-119::gfp* worms (day-3 adult) grown on control or *cyd-1* RNAi. White arrows mark normal eggs that do not show GFP expression, while yellow arrows mark endomitotic oocytes that express GFP (K). Quantification of *unc-119*::GFP expression positive endomitotic oocytes (L). Average of four biological replicates (n ≥ 25 for each experiment). Unpaired *t*-test with Welch's correction. Scale bars: 20 μm. Error bars are s.d. Experiments were performed at 20˚C. Source data are provided in S1 Table.

stage does not lead to poor oocyte quality (S1I Fig) or endomitotic oocytes (S1J Fig), indicating a role of CYD-1 during development, that may later affect the oocyte quality in adulthood. Results here demonstrate that CYD-1 is required to maintain optimal oocyte quality in wild-type worms.

## CYD-1 is required in the somatic gonad (uterus) for oocyte quality maintenance

Previously, the IIS and the TGF-β signaling in the somatic tissues (hypodermis, muscle and intestine) have been shown to regulate reproductive aging [18]. The somatic tissues crosstalk with the germline, not only to regulate aging [62–64] but also to influence germline development [65] and the oocyte quality [18]. Keeping this in mind, we wanted to test the tissue-specific requirement of CYD-1 in the regulation of oocyte quality. For this, we employed a tissue-specific RNAi system where in the *rde-1* (gene encoding an Argonaute protein) mutant background, a tissue-specific promoter is used to drive RDE-1 expression, providing an elegant system to selectively knock-down a gene in that specific tissue alone [66]. Surprisingly, in contrast to the whole-body RNAi of *cyd-1*, KD of *cyd-1* in the germ cells [using a transgenic line where *sun-1* promoter drives the expression of RDE-1 only in the germ cells of the *rde-1* mutant [67]] alone was unable to deteriorate the oocyte quality (**Fig 2A and 2B**), increase the percentage of endomitotic oocytes (**Fig 2C and 2D**) or affect the brood size (**S2A Fig**). Interestingly, somatic tissue-specific KD using a *ppw-1* [a PAZ/PIWI protein required for efficient germline RNAi [68]] mutant resulted in poor oocyte quality (**Fig 2A and 2B**) with increased emos (**Fig 2C and 2D**) and a concomitant reduction in brood size (**S2B Fig**). The germline and somatic RNAi specificity of these mutants was validated by *egg-5* RNAi [69] (which leads to the laying of dead eggs when RNAi is functional in the germline) and *dpy-7* RNAi [70] (which leads to dumpy phenotype when RNAi is functional in the soma) (**S2C Fig**).

To further investigate the somatic tissue where CYD-1 may function to influence the germline quality, we used the tissue-specific strains in the *rrf-3* mutant background. Intestine or muscle-specific KD of *cyd-1* resulted in no changes in the oocyte morphology (**S2D and S2E Fig**). Surprisingly, uterine-specific KD of *cyd-1* [using a transgenic line where *fos1a* promoter drives the expression of RDE-1 only in the uterine cells of the *rde-1* mutant [71]] led to compromised oocyte quality (**Fig 2E and 2F**) with an increase in the emos (**S2F and S2G Fig**) and a reduction in the total progeny number (**S2H Fig**). The strain was confirmed to have RNAi machinery specifically in the uterine cells as it showed a protruded vulva phenotype upon *egl-43* KD [EGL-43 is essential for cell fate specification and development of vulva and uterus [72]] and no dead eggs or dumpy phenotype on *egg-5* and *dpy-7* RNAi, respectively (**S2I Fig**).

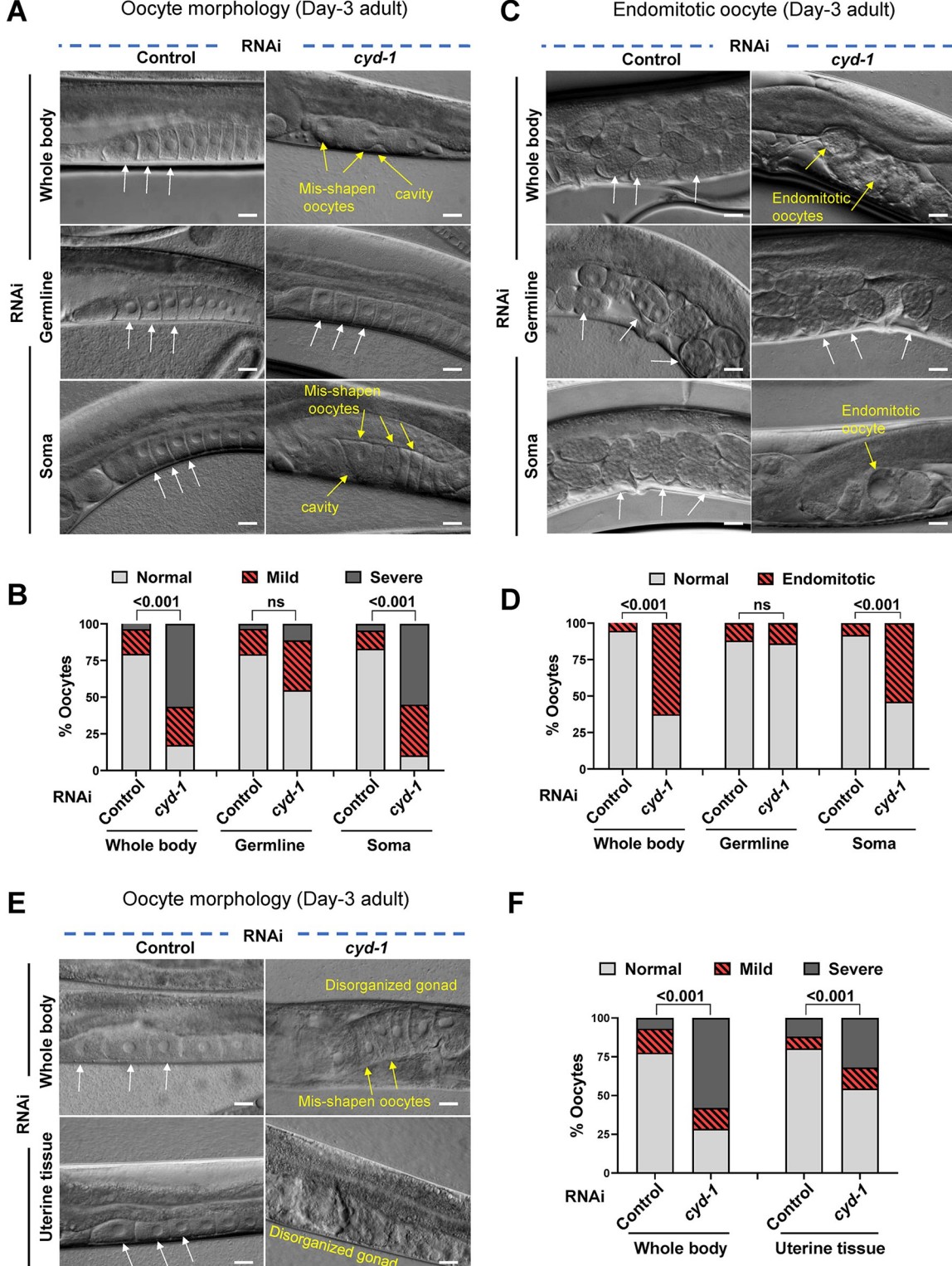

**Fig 2. CYD-1 regulates oocyte quality cell non-autonomously from the somatic gonad (uterus). (A,B)** DIC images showing oocyte morphology of WT, *rde-1(mkc36);sun-1p::rde-1* (germline-specific RNAi) and *ppw-1(pk1425)* (soma-specific RNAi) (day-3 adult) grown on control or *cyd-1* RNAi. White arrows- normal oocytes, yellow arrows- oocytes with abnormalities (A). Quantification of oocyte quality score (B). The quality of oocytes was categorized as normal, mild or severe based on their morphology (cavities, shape, organization) as shown in S1G Fig. Combined data from three biological replicates (n ≥ 29) is plotted. Chi-square analysis was used to

compare between groups. **(C,D)** DIC images showing the presence of eggs or endomitotic oocytes (emos) in the uterus of WT, *rde-1 (mkc36);sun-1p::rde-1* (germline-specific RNAi) and *ppw-1(pk1425)* (soma-specific RNAi) (day-3 adult) grown on control or *cyd-1* RNAi. White arrows mark normal eggs while yellow arrows mark endomitotic oocytes (emos) (C). Quantification for endomitotic oocytes (D). Combined data from three biological replicates (n ≥ 25) is plotted. Chi-square analysis was used to compare between groups. **(E,F)** DIC images showing oocyte morphology of *rrf-3(pk1426)* (whole body RNAi) or *rrf-3(pk1426);rde-1(ne219);fos-1ap::rde-1 (genomic)* (uterine tissue-specific RNAi) worms (day-3 adult) grown on control or *cyd-1* RNAi. White arrows mark normal oocytes while yellow arrows mark oocytes with severe abnormalities (E). Oocyte quality scores (based on morphology as shown in S1G Fig) (F). Combined data from three (whole body KD) and four (uterine specific KD) biological replicates (n ≥ 58) are plotted. Chi-square analysis was used to compare between groups. Scale bars: 20 μm. Experiments were performed at 20˚C. Source data are provided in S1 Table.

Thus, CYD-1 is important in the somatic gonad (uterine tissue) for oocyte quality maintenance in the WT germline. This exemplifies a soma-to-germline crosstalk that ensures the quality of the germline.

## Depletion of *cyd-1* leads to a DAF-16a isoform-dependent pachytene arrest in *daf-2*

While the RNAi-mediated KD of *cyd-1* accelerates the decline in oocyte quality in WT, RNAi depletion of *cyd-1* under low insulin signaling (*daf-2* worms) leads to complete sterility, as mentioned earlier (**Figs 3A–3C and S1A**). Interestingly, similar to stress resistance [11–14] and longevity [73,74], this germline arrest also was mediated by the downstream FOXO TF DAF-16, such that the fertility is rescued in *the daf-16;daf-2;cyd-1* RNAi worms (**Fig 3A–3C**). To determine the effect of *cyd-1* KD on germline development of *daf-2*, the gonads of day-1 adults were stained with DAPI. Interestingly, the *cyd-1* KD did not lead to a reduction in the mitotic zone germ cell number. Importantly, however, no oocytes were formed upon *cyd-1* KD in *daf-2* and the oocyte formation was restored in the *daf-16;daf-2* double mutants (**Fig 3D and 3E**). Upon quantification of the germ cell numbers, we found that *cyd-1* KD led to small but significant changes in the transition zone germ cells, but drastically reduced the pachytene germ cell number in *daf-2*; this was restored in *daf-16;daf-2* (**Fig 3D and 3E**). We measured *cyd-1* mRNA levels and confirmed similar efficiency of *cyd-1* KD in *daf-2* and *daf-16;daf-2* worms, implying that the germline arrest is not due to a differential *cyd-1* RNAi efficiency (**S3A Fig**). We further speculated if KD of *cyd-1* in *daf-2* worms resulted in critically low levels of CYD-1 levels below a presumptive threshold, thereby causing sterility. To rule out this, we depleted *cyd-1* (by RNAi) in a balanced *cyd-1* mutant, *cyd-1(he112)/+* that possesses a truncated (C-terminal) CYD-1 protein due to the presence of a nonsense mutation [75]. Interestingly, we observed deterioration of oocyte quality (**S3B and S3C Fig**) and increased formation of endomitotic oocytes (**S3D and S3E Fig**), but no sterility, implying the arrest is specific to *daf-2*. Thus, upon depletion of *cyd-1* in *daf-2*, activated DAF-16 plays a decisive role in arresting the germ cells at the pachytene stage of meiosis and in preventing oocyte formation.

Since the RNAi depletion of *cyd-1* phenocopies the sterility (pachytene arrest) due to *cdk-12* KD in *daf-2*, we asked if similar to *cdk-12* depletion, *cyd-1* KD too weakened the DNA damage response (DDR). For this, we first checked for the mRNA levels of genes involved in the DDR. Wild-type late-L4 staged worms grown on control or *cyd-1* RNAi showed no down-regulation of DDR genes on *cyd-1* KD (**S3F Fig**). Additionally, both control and *cyd-1* RNAi-fed WT worms showed a similar percentage of egg hatching post gamma exposure at late-L4 stage (**S3G Fig**). This hinted towards the existence of different mechanisms underlying the sterility upon *cyd-1* and *cdk-12* KD in *daf-2*.

One of the causes of sterility in *daf-2;cyd-1* RNAi worms could be elevated levels of germ cell apoptosis. For measuring apoptosis, we employed the CED-1::GFP transgenic strain as described above. Elevated levels of apoptosis were observed for both *daf-2* and *daf-16;daf-2*

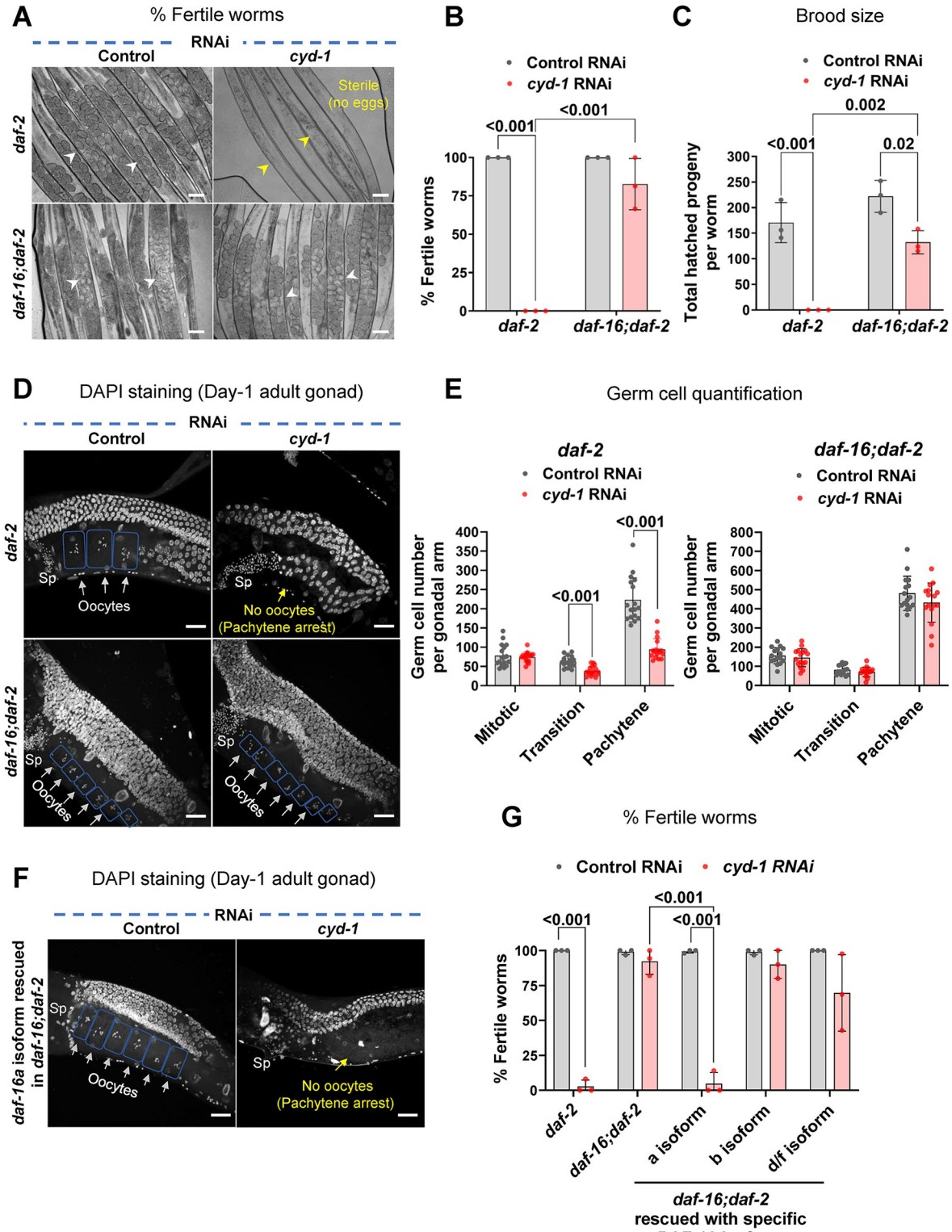

**Fig 3. Depletion of *cyd-1* leads to FOXO/DAF-16-dependent germline arrest under low insulin signaling. (A,B)** Representative images showing that *cyd-1* RNAi results in sterility in *daf-2(e1370)* worms that are rescued in *daf-16(mgdf50);daf-2(e1370)*. White arrowheads show eggs while yellow arrowheads show gonads with no eggs (sterile worms) (A). The percentage of fertile worms (B). Average of three biological replicates (n ≥ 30 for each experiment). Two-way ANOVA-Tukey's multiple comparisons test. **(C)** The total number of hatched progenies in *daf-2(e1370)* and *daf-16(mgdf50);daf-2(e1370)* worms grown on control or *cyd-1* RNAi. Average of three biological repeats.

Two-way ANOVA-Tukey's multiple comparisons test. **(D,E)** Representative fluorescent images of DAPI-stained gonads of *daf-2(e1370)* and *daf-16(mgdf50);daf-2(e1370)* (day-1 adult) worms grown on control or *cyd-1* RNAi. Oocytes are boxed for clarity. White arrows point towards oocytes while yellow shows the absence of oocytes. Sp denotes sperms (D). Quantification of DAPI-stained germ cell nuclei. n = 17 (*daf-2*); n = 16 (*daf-16;daf-2*). (E) Each point represents the number of mitotic (MT), transition (TS) or pachytene zone (PZ) cells. Unpaired *t*-test with Welch's correction. **(F,G)** Isoform requirement of DAF-16 to mediate germline arrest in *daf-2(e1370)*. Representative fluorescent images of DAPI-stained germ line of *daf-16(mgdf50);daf-2(e1370);daf-16a(+)* worms grown on control or *cyd-1* RNAi. Oocytes are boxed for clarity. White arrows point towards oocytes while yellow shows the absence of oocytes. Sp denotes sperms (F). Percentage of the fertile worms (G). The *cyd-1* was knocked down in *daf-2(e1370)*, *daf-16(mgdf50);daf-2(e1370)* as well as in strains where different DAF-16 isoforms were rescued in *daf-16(mgdf50);daf-2(e1370)*. Average of three biological replicates (n ≥ 30 for each replicate). Two-way ANOVA-Tukey's multiple comparisons test. Scale bars: 20 μm. Error bars are s.d. Experiments were performed at 20˚C. Source data are provided in S1 Table.

worms grown on *cyd-1* RNAi (**S3H and S3I Fig**), implying that the sterility in *daf-2* upon *cyd-1* KD may not be solely due to an increase in apoptosis. We wondered if the oocytes that bypass the pachytene arrest in *daf-16;daf-2;cyd-1* RNAi worms would also exhibit poor quality. Indeed, we observed that *cyd-1* KD in *daf-16;daf-2* led to severe defects in oocyte quality (**S4A and S4B Fig**) as well as an increased occurrence of emo in the uterus of these animals (**S4C and S4D Fig**).

Different isoforms of DAF-16 exist due to alternative splicing events as well as alternate promoter usage. The different isoforms have distinct tissue-specific expression profiles and the long life span of *daf-2* depends on each isoform to a different extent [76]. We next asked which isoform of DAF-16 was necessary for inducing arrest in *daf-2* following KD of *cyd-1*. To address this, we used transgenic strains, wherein, different isoforms of DAF-16 were rescued in the *daf-16;daf-2* double mutants [76]. Interestingly, we found that the *cyd-1* KD in the *daf-16;daf-2;daf-16a(+)* strain led to sterility comparable to the *daf-2* (**Fig 3F and 3G**), signifying a predominant role of DAF-16a isoform in mediating this germline pachytene arrest.

Additionally, the sterility due to *cyd-1* KD in *daf-2* involves the canonical IIS pathway components as a similar arrest at the pachytene stage of meiosis was observed upon *cyd-1* KD in *age-1(hx546)* [mutant in mammalian PI3K ortholog [77, 78]] and *pdk-1(sa680)* [mutant in mammalian PDK ortholog [79]] (**S4E and S4F Fig**). Reducing the levels of the canonical binding partner of CYD-1, CDK-4 [80] also led to sterility in *daf-2;rrf-3* (hypersensitive RNAi background in *daf-2*) (**S4G Fig**), hinting towards a CDK-4-dependent function of CYD-1 in oogenesis.

## Depletion of *cyd-1* in the somatic gonad of *daf-2* results in germline arrest at the pachytene stage

Since we found a role for somatic CYD-1 in regulating WT germline quality assurance, we asked if somatic *cyd-1* KD would lead to germline arrest in *daf-2*. We first checked if the germline arrest was mediated cell-autonomously by *cyd-1* depletion in the germ cells. Interestingly, *cyd-1* depletion in the germ cells alone did not lead to sterility in *daf-2* (**Fig 4A and 4B**). However, *cyd-1* KD specifically in the uterine cells *per se* resulted in the germ cell arrest at the pachytene stage (**Fig 4A and 4B**) and also caused a reduction in the pachytene germ cell number (**Fig 4C**). This somatic requirement of CYD-1 in germline arrest reiterates a non-canonical role of CYD-1 in oogenesis. Depletion of *cyd-1* in other somatic tissues like muscle, neurons, intestine, hypodermis, Distal Tip Cell (DTC) and the vulva did not cause sterility in *daf-2* (**S5A Fig**). Interestingly, we observed vulval defects (protruded vulva/vulvaless) upon *cyd-1* KD in *daf-2* while *daf-16;daf-2* showed no such defects (**S5B and S5C Fig**).

The egg-laying apparatus consists of the uterus, the uterine muscles, the vulva, the vulval muscles, and the egg-laying neurons. The uterus and vulva are connected by the uterine seam cell [81] (**Fig 4D**). The vulva formation starts at the L3 larval stage and vulval cell-divisions are

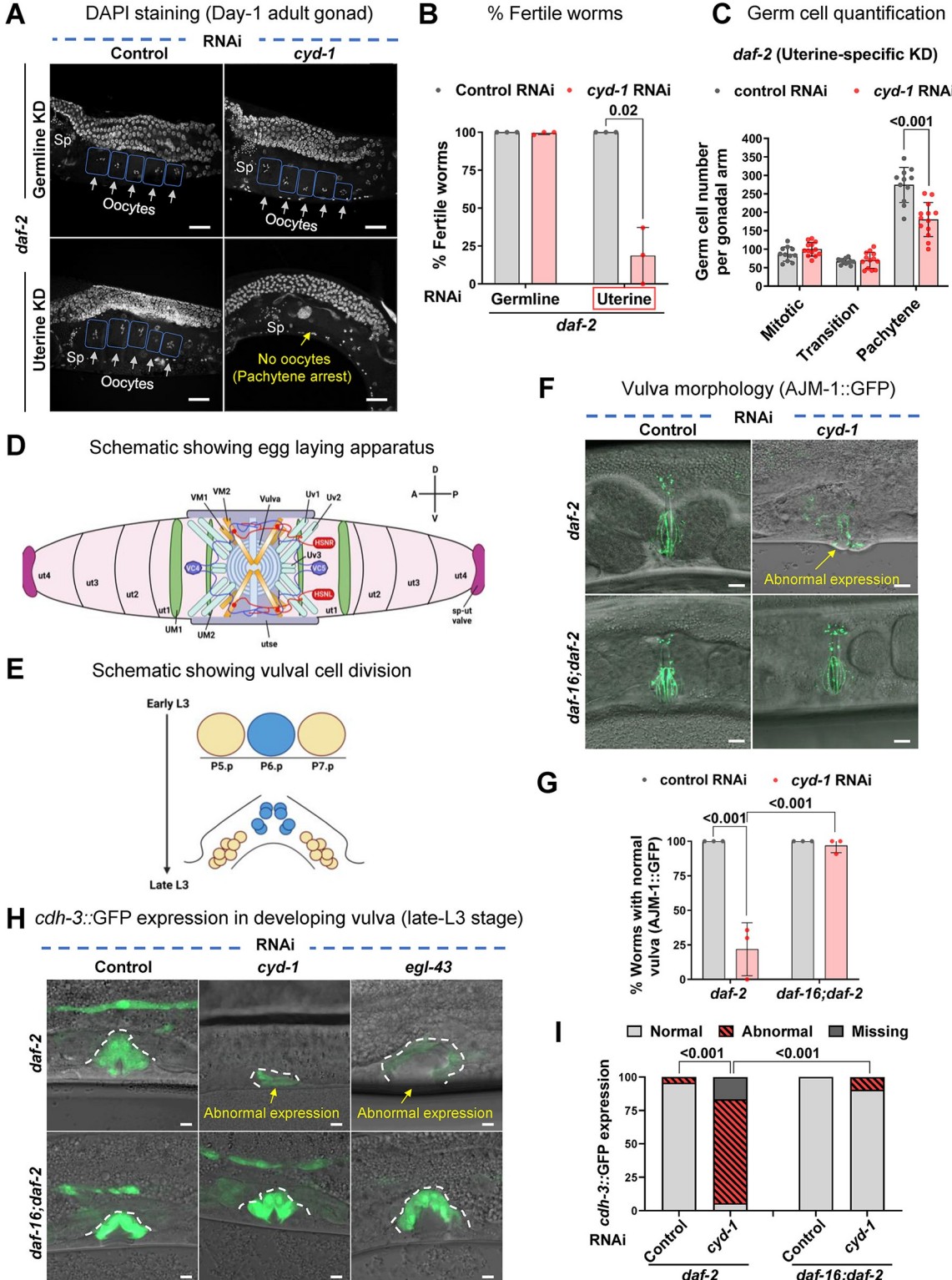

**Fig 4. Depletion of *cyd-1* in the somatic gonad (uterus) alone is sufficient to cause germline arrest under low insulin signaling.**
**(A,B)** Representative images of DAPI-stained gonads showing no pachytene arrest in day-1 adult *daf-2(e1370);rde-1(mkc36);sun-1p::rde-1* (germline-specific RNAi) worms upon *cyd-1* KD. However, pachytene arrest ensues upon *cyd-1* KD in *daf-2(e1370);rrf-3 (pk1426);rde-1(ne219);fos-1ap::rde-1* (uterine tissue-specific RNAi). Oocytes are boxed for clarity. White arrows point towards oocytes while yellow shows the absence of oocytes. Sp denotes sperms. Scale bar 20 μm (A). The percentage of fertile worms upon germline-

specific and uterine-specific *cyd-1* KD in *daf-2(e1370)* (B). Average of three biological replicates (n ≥ 25 per condition for each experiment). Unpaired *t*-test with Welch's correction. (**C**) Quantification of DAPI-stained *daf-2(e1370);rrf-3(pk1426);rde-1(ne219);fos-1ap::rde-1* (uterine tissue-specific RNAi) (day-1 adult) germ cell nuclei of different stages (mitotic (MT), transition (TS) or pachytene zones (PZ)); n = 11. Each point represents the number of mitotic (MT), transition (TS) or pachytene zone (PZ) cells. Unpaired *t*-test with Welch's correction. (**D**) Schematic showing the egg laying apparatus of an adult *C. elegans* hermaphrodite. The egg-laying apparatus consists of the uterus, the uterine muscles, the vulva, the vulval muscles, and the egg-laying neurons. A stack of seven nonequivalent epithelial toroids (rings) forms the vulva. The anterior and posterior lobes of uterus each comprises of four uterine toroid epithelial cells (ut1–ut4). Adherens junctions connect the neighboring uterine toroids or vulval toroids, respectively. The uterine seam cell (utse) is an H shaped cell that connects the uterus to seam cells and thus holds the uterus in place. uv1-3 are interfacial cells between uterus and vulva (uv1 is a neuroendocrine cell). The process of egg-laying is made possible through the contraction of sex muscles, namely the vulval muscles (VM1,2) connected to the vulva lips and the uterine muscles (UM1,2) encircling the uterus. The activity of these muscles is under the regulation of motor neurons, specifically VCn (VC1-6) and HSNL/R, which form synapses with each other and with the vulval muscle arms. A = anterior, P = posterior, D = dorsal, V = ventral. Adapted from Wormbook (doi:10.3908/wormatlas.1.24). (**E**) Schematic showing the vulval cell divisions. The vulval precursor cells (VPCs) P5.p, P6.p and P7.p are induced to divide in the early L3 stage. Three rounds of cell division yield a total of 22 vulval cells by the end of L3 stage. The adult vulva is then formed through subsequent patterning and morphogenesis processes. Schematics in Fig 4D and 4E were created using BioRender.com. (**F,G**) Representative fluorescent and DIC merged images of gonads showing AJM-1::GFP (marks cell junctions) expression in *daf-2(e1370)* and *daf-16(mgdf50);daf-2(e1370)* worms (day-1 adult) grown on control or *cyd-1* RNAi. Yellow arrows point towards abnormal expression. Scale bar 10 μm (F). Quantification of the normal or abnormal expression of AJM-1::GFP (G). Average of three biological replicates (n ≥ 24 for each experiment). Two-way ANOVA-Tukey's multiple comparisons test. (**H,I**) Representative fluorescent and DIC merged images of gonads showing *cdh-3*::GFP expression in *daf-2(e1370)* and *daf-16(mgdf50);daf-2(e1370)* worms (late-L3 stage) on control, *cyd-1* or *egl-43* RNAi. Yellow arrows point towards abnormal expression. Scale bar 5 μm (H). Quantification for normal, reduced or missing *cdh-3*::GFP expression (I). Combined data from three biological replicates (n ≥ 36) is plotted. Chi-square analysis was used to compare between groups. Error bars are s.d. Experiments were performed at 20˚C. Source data are provided in S1 Table.

completed by the late-L3 stage (**Fig 4E**). We investigated if *cyd-1* KD caused defects in the vulva-uterine morphology. We visualized the vulval muscle architecture by Phalloidin staining (that marks the F-actin) that revealed a distorted vulva muscle structure following the depletion of *cyd-1* in *daf-2* (**S5D Fig**). To further dissect the underlying cause of the vulval defects, we employed the AJM-1::GFP-expressing transgenic strain (which marks the adherens junction) [82]. Vulval development involves the formation of a tubular structure from only 22 epithelial cells. It encompasses an array of steps including vulval cell induction and differentiation, cell division, cell rearrangements, invagination, toroid formation and cell-cell fusion and attachment with other tissues (hypodermis, muscle, uterus) [83]. Normally, in the vulva, the AJM-1::GFP marks the symmetrical toroid; however, upon *cyd-1* depletion in *daf-2*, the AJM-1::GFP was mis-localized and the expression was diminished. In contrast, *cyd-1* KD in *daf-16;daf-2* worms did not exhibit such abnormalities (**Fig 4F and 4G**). This highlights the crucial role of activated DAF-16 in causing vulval impairments consequent to *cyd-1* KD. Next, we asked if these defects in vulva were due to impairment in the vulva development (vulval cell divisions or cell fate acquisition). We observed the vulval precursor cell (VPC) fate of late-L3 staged worms using the transgenic reporter strain, *cdh-3*::GFP. CDH-3 belongs to the cadherin family of proteins and is expressed in the cells of the egg laying apparatus (vulva, uterine cells, HSN neurons) during development [84]. In *daf-2* worms fed control RNAi, we observed the correct expression of *cdh-3*::GFP in vulval cells, however, the expression was abnormal or absent upon *cyd-1* RNAi, suggesting incomplete or aberrant vulval development. Notably, the *daf-16;daf-2;cyd-1* RNAi animals exhibited a restoration of normal *cdh-3*::GFP expression in vulval cells (**Fig 4H and 4I**). Furthermore, we utilized the depletion of the *egl-43* gene [EGL-43 is essential for vulval morphogenesis [72,85]] as a control, and observed analogous abnormalities in *cdh-3*::GFP expression (**Fig 4H**). Interestingly, in this context as well, we observed a DAF-16-dependence such that the *cdh-3*::GFP expression was restored in the *daf-16;daf-2* worms (**Fig 4H**). Additionally, *egl-43* KD in *daf-2* resulted in endomitotic oocytes and sterility, which was partially rescued in the *daf-16;daf-2* worms (**S5E and S5F Fig**). These observations allude to the possibility that activated DAF-16 may amplify deficiencies in the egg-laying apparatus, potentially hindering normal oogenesis.

## Depletion of *cyd-1* results in DAF-16-dependent defects in the somatic gonad of *daf-2* worms

To gain deeper insights into the changes underlying the germline arrest due to *cyd-1* KD in *daf-2*, we performed transcriptomics analysis of the late-L4 staged WT and *daf-2* worms with *cyd-1* KD only in the somatic gonad [using a transgenic strain where *unc-62* promoter drives the expression of RDE-1 only in the egg-laying apparatus in the *rde-1* mutant [86]]. Interestingly, the genes important for cell cycle/cell-fate, oogenesis and reproduction were significantly downregulated in *daf-2;cyd-1* RNAi worms (**S6A Fig**), but not in *cyd-1* KD in the WT worms (**S6B Fig**). Importantly, the genes involved in vulva-uterine [*lin-11* [87], *lim-6* [88], *unc-59* [89]] and sheath cell development [*lim-7* [90]] were also down-regulated in the *daf-2* gonad-specific *cyd-1* KD (**Fig 5A** and S2 Table) but not in WT (**S6C Fig** and S3 Table). The gonadal sheath cells are a crucial component of the worm somatic gonad that plays an important role in pachytene exit and oogenesis [56, 91–93]. The 5 pairs of sheath cells surround each gonadal arm (**Fig 5B**). We tested if *cyd-1* KD could also lead to defects in the sheath cells. For this, we used the sheath cell marker strain *lim-7p*::GFP [92] [LIM-7 is one of the LIM-homeodomain transcription factor [90]]. Interestingly, we found that the *cyd-1* KD in *daf-2* led to the complete abrogation of the *lim-7p*::GFP expression, which was restored in the *daf-16;daf-2* (**Fig 5C**). We stained the gonads of day-1 adults with Phalloidin (labels F-actin) to investigate potential alterations in sheath cell structure in *daf-2;cyd-1* RNAi worms. Strikingly, we observed a significant disruption in the cytoskeletal organization of sheath cells in *daf-2;cyd-1* RNAi worms, which was ameliorated in *daf-16;daf-2* worms subjected to *cyd-1* RNAi (**Figs 5D and S6D**). We inquired if knocking down genes essential for sheath cell development (e.g., *lim-7*, *lim-6)* would also lead to germline arrest in *daf-2* but found no such sterility (**S7 Fig**). However, similar to *cyd-1* RNAi, we observed a DAF-16-dependent pachytene arrest and sterility (**Fig 5E**) and repression of *lim-7p*::GFP expression (**S6E and S6F Fig**) upon KD of *sys-1*. SYS-1/β-catenin transcription factor acts downstream of the WNT signaling pathway and is known to play a key role in the development of the somatic gonad [94, 95]. This suggests that activated DAF-16 may amplify somatic gonad defects upon perceiving potential threats in the somatic reproductive tissue (such as CYD-1 or SYS-1 depletion), thereby halting oogenesis progression to ensure poor quality eggs are not laid.

Next, we investigated if *cyd-1* RNAi can alter the expression of another adult hermaphrodite gonad-specific marker (FKH-6::GFP). In the hermaphrodite gonad, the forkhead transcription factor, FKH-6 expresses in the adult spermatheca and is important for gonadal cell differentiation [96]. Similar to *lim-7p*::GFP expression, we found a DAF-16-dependent suppression of FKH-6::GFP expression upon *cyd-1* depletion in *daf-2* (**S6G and S6H Fig**). Interestingly, while the WT worms showed no obvious structural defects in the sheath cell upon *cyd-1* depletion (**S6I and S6J Fig**), *lim-7p*::GFP expression was significantly reduced in day-3 *cyd-1* KD WT worms compared to age-matched control worms (no difference in day-1 adults) (**Fig 5F**).

Overall, these data imply that *cyd-1* depletion elicits a DAF-16-dependent distortion of sheath cell structure and suppression of hermaphrodite gonad marker expression in *daf-2*. The amplification of somatic gonadal abnormalities when activated DAF-16 is present may, in turn, influence oogenesis.

## Lowering CYD-1 levels may cause defective sperm-to-oocyte fate switch in *daf-2*

In the hermaphrodites, a fixed number of sperms are formed in the L4 larval stage, after which the fate is switched from sperm to oocyte and adult worms make only oocytes from then onwards [97]. Upon *cyd-1* KD in *daf-2*, we observed only sperms but no oocytes in the adult

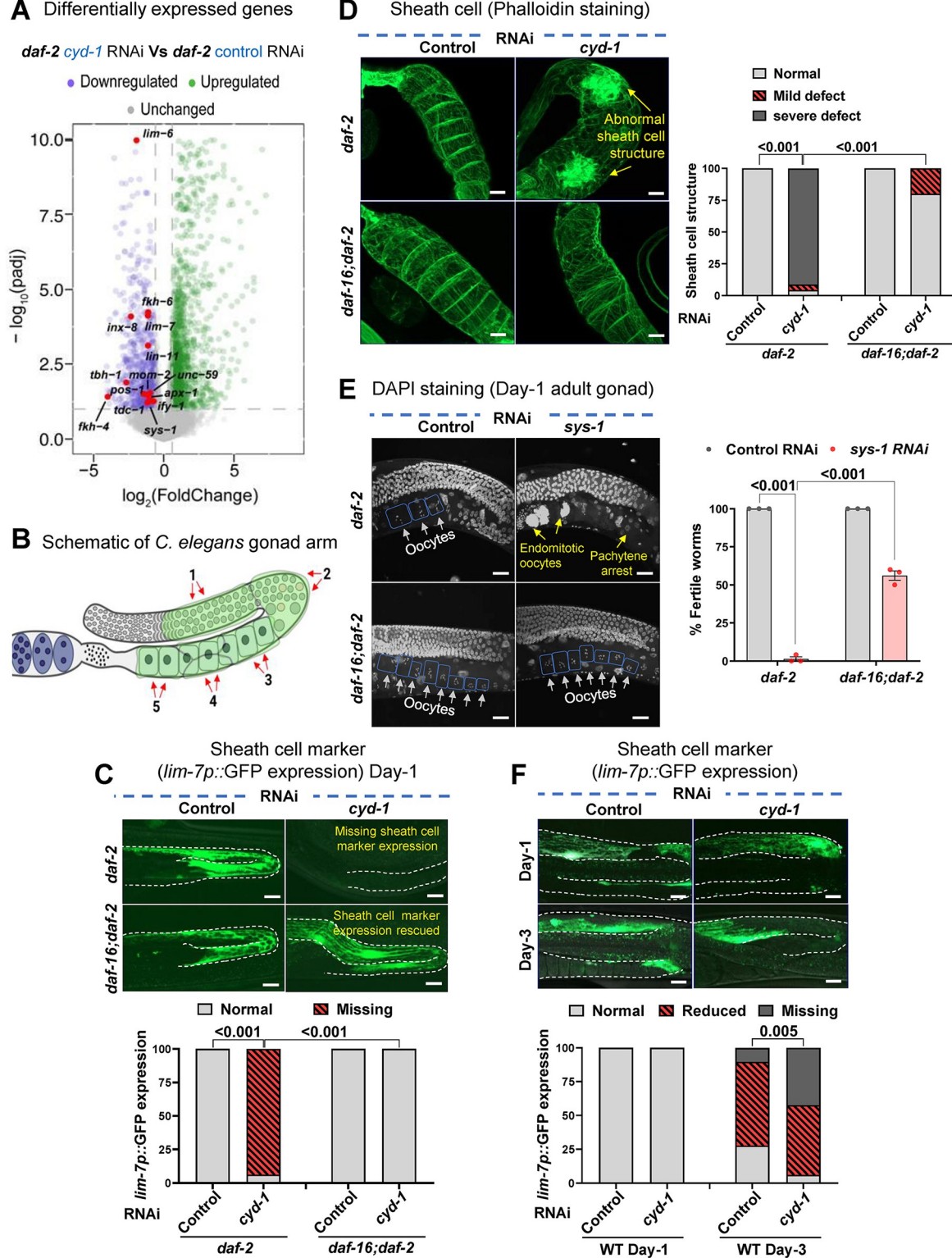

**Fig 5. Depletion of *cyd-1* under low insulin signaling leads to DAF-16-dependent defects in sheath cells. (A)** Volcano plot showing the magnitude [log₂(FC)] and significance [−log₁₀(P value)] of the genes that are differentially expressed in L4-staged *daf-2(e1370);rrf-3 (pk1426);rde-1(ne219);unc-62p::rde-1(genomic)* worms, grown on control or *cyd-1* RNAi. Y axis [−log₁₀(P value)] is restricted to a value of 10. **(B)** Schematic showing one of the two gonad arms of an adult *C. elegans* hermaphrodite. Sheath cells are colored green. Schematic in Fig 5B were created using BioRender.com. **(C)** Representative fluorescent and DIC merged images of gonads showing *lim-7p*::GFP (that

marks the sheath cells) expression in *daf-2(e1370)* and *daf-16(mgdf50);daf-2(e1370)* worms (day-1 adult) grown on control and *cyd-1* RNAi. The gonadal arm is outlined for clarity. Scale bar 20 μm. Quantification of the normal or missing expression of *lim-7p*::GFP is given below the figure. Combined data from three biological replicates (n ≥ 65) is plotted. Chi-square analysis was performed to compare between groups. **(D)** Representative fluorescent images of Phalloidin-stained (that marks the F-actin) gonads of *daf-2(e1370)* and *daf-16 (mgdf50);daf-2(e1370)* worms (day-1 adult) grown on control and *cyd-1* RNAi. Arrows point towards defects in the sheath cells. Scale bar 10 μm. Quantification of the normal or defective sheath cell structure (as per scoring scheme in S6D Fig) is shown on the right. Combined data from three biological replicates (n ≥ 12) is plotted. Chi-square analysis was performed to compare between groups. **(E)** Representative fluorescent images of DAPI-stained gonads of *daf-2(e1370)* and *daf-16(mgdf50);daf-2(e1370)* (day-1 adult) worms grown on control or *sys-1* RNAi. Oocytes are boxed for clarity. White arrows point towards oocytes while yellow shows endomitotic oocytes or lack of oocytes. Scale bar 20 μm. The percentage of fertile worms is shown on the right. Average of three biological replicates (n ≥ 25 per condition for each experiment). Two-way ANOVA-Tukey's multiple comparisons test. **(F)** Representative fluorescent images of gonads showing *lim-7p*::GFP (that marks the sheath cells) expression in WT worms (day-1 and day-3 adult) grown on control and *cyd-1* RNAi. Arrows point towards defective/missing *lim-7p*::GFP expression in the sheath cells. The gonadal arm is outlined for clarity. Scale bar 20 μm. Quantification of the normal/defective or missing *lim-7*::GFP expression is given below the figure. Combined data from three biological replicates (n ≥ 39) is plotted. Chi-square analysis was performed to compare between groups. Error bars are s.d. Experiments were performed at 20˚C. Source data are provided in S1 Table.

hermaphrodite worms (**Fig 6A**). Moreover, we showed above that lowering CYD-1 levels in *daf-2* suppresses the hermaphrodite gonad-specific marker expression (**Fig 5C, S6G and S6H**). We supposed that either the germline has been masculinized (male mode of germline: continuous sperm production) or there may be a failure in the sperm-to-oocyte fate switch. However, we did not observe any ectopic expression of the male gonad marker (*K09C8.2p*:: GFP) [98] in *daf-2;cyd-1* RNAi worms (**S8 Fig**). Also, the sperm number remained unchanged in the *cyd-1*-depleted *daf-2* young adult worms in comparison to control RNAi grown worms (**Fig 6B**) and the sperm number did not increase on day-2 or day-3 of adulthood, indicating that the germline had not been masculinized (**Fig 6C**). Alternatively, *cyd-1* depletion in *daf-2* may cause a defect in the sperm-to-oocyte cell fate switch and a consequent arrest of oogenesis at the pachytene stage. In line with this assumption, the depletion of *cyd-1* led to a DAF-16-dependent downregulation of genes essential for the sperm-to-oocyte fate switch (**Fig 6D and 6E**). Interestingly, we also observed the transcripts of these sperm-to-oocyte switch genes to be downregulated upon KD of *cyd-1* only in the uterine cells of the *daf-2* (**Fig 6F**). Since the proximal-most sheath cell pairs play an important role in the pachytene exit [56,65], the deformation of sheath cells in the *daf-2;cyd-1* RNAi worms may be the driver for the defective sperm-to-oocyte fate switch, leading to sterility in these animals.

## Discussion

The role of cyclin D/CDK4 in the progression of G1 to S phase of the cell cycle is well known. Research in the past decade has shed light on several non-cell cycle functions of cyclin D, including its role in adipogenesis [45,46], muscle differentiation [47,48], anoikis [99], DNA double-strand break (DSB) repair [49] and transcriptional regulation [50–52], many of them independent of its binding partner, CDK4 [49,100–102]. Here, we decipher a novel role of *C. elegans* cyclin D/CYD-1 in preserving the quality of oocytes. Depletion of *cyd-1* in wild-type nematodes led to misshapen oocytes with increased cavities between them and disruption of gonadal organization, akin to that of aged worms. Concurrently, endomitotic oocytes increased in the germline of these worms, with a decline in egg hatchability (**Fig 7**). These findings underscore the significance of CYD-1 in the regulation of reproductive fidelity. At an organismal level, environmental cues are integrated by the nutrient sensing pathways that often coordinate with cell-cycle regulators to influence reproductive cell fate [103–106]. Lowering a key nutrient-sensing pathway, the worm IIS, causes activation of the downstream FOXO/DAF-16 TF leading to improved oocyte quality and delayed reproductive senescence [17,18]. In mammals too, FOXO3a has been shown to maintain the ovarian reserve [35,107],

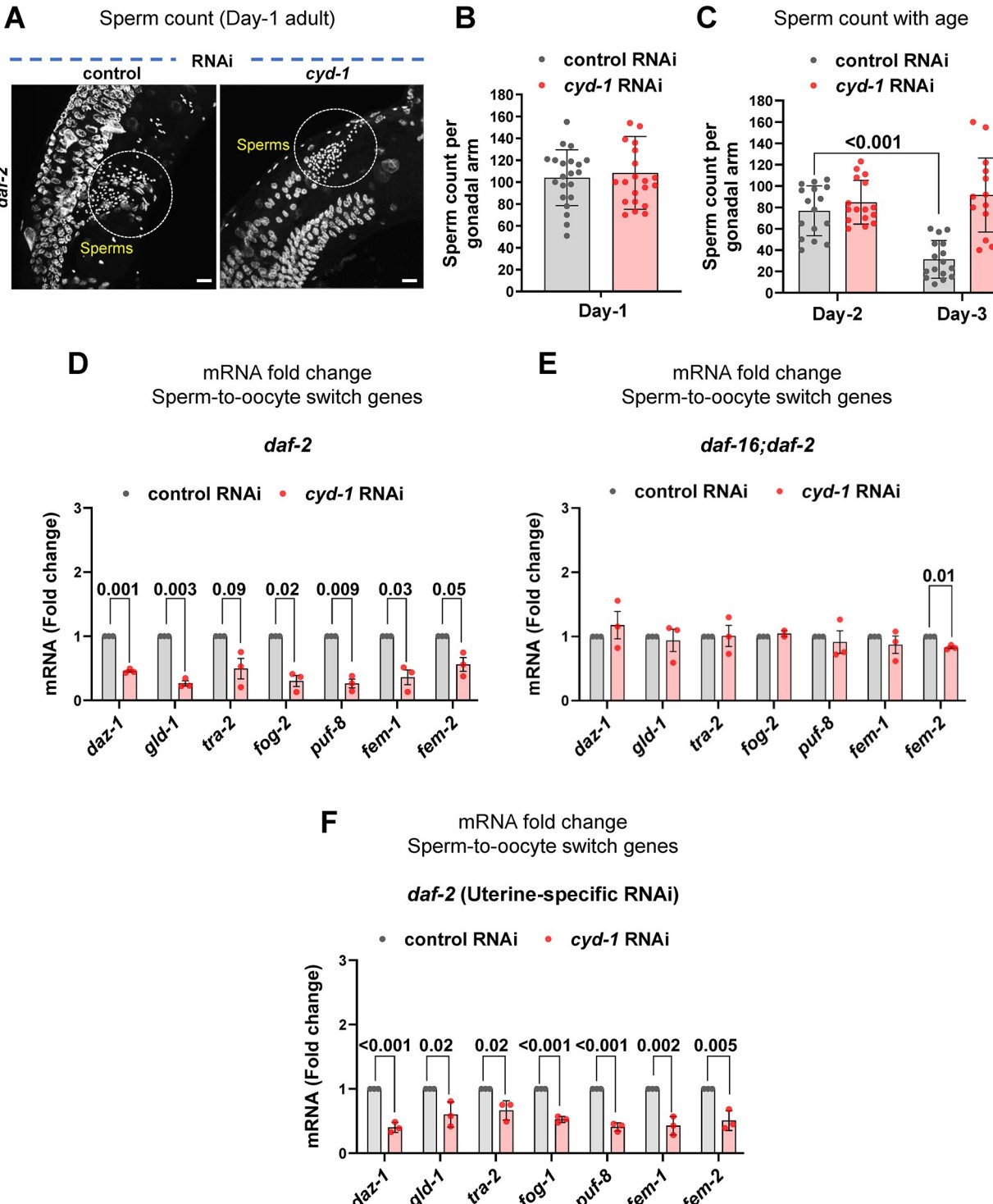

**Fig 6. A defective sperm-to-oocyte switch may underlie germline arrest upon *cyd-1* KD under low insulin signaling. (A,B)** Representative fluorescent images of DAPI stained *daf-2(e1370)* worms (early day-1 adult) grown on control and *cyd-1* RNAi. Sperms are encircled for clarity. Scale bar 10 μm (A). Quantification of the sperm count (n = 21) (B). Each point represents the number of sperms per gonadal arm per worm. Unpaired *t*-test with Welch's correction. **(C)** Quantification of the sperm count in *daf-2(e1370)* worms (day-2 and day-3 adults) grown on control and *cyd-1* RNAi. Each point represents the number of sperms per gonadal arm per worm (n = 16). Two-way ANOVA-Tukey's multiple comparisons test. **(D)** Quantitative RT-PCR analysis of sperm-to-oocyte switch genes in *daf-2(e1370)* worms (late-L4 stage) grown on control or *cyd-1* RNAi. Expression levels were normalized to *actin*. Average of three biological replicates are shown. Unpaired *t*-test with Welch's correction. **(E)** Quantitative RT-PCR analysis of sperm-to-oocyte switch genes in *daf-16(mgdf50);daf-2(e1370)* worms (late-L4 stage) grown on control or *cyd-1* RNAi. Expression levels

were normalized to *actin*. Average of three biological replicates are shown. Unpaired *t*-test with Welch's correction. (**F**) Quantitative RT-PCR analysis of sperm-to-oocyte switch genes in *daf-2(e1370);rrf-3(pk1426);rde-1(ne219);fos-1ap::rde-1* (uterine tissue-specific RNAi) worms (late-L4 stage) grown on control or *cyd-1* RNAi. Expression levels were normalized to *actin*. Average of three biological replicates are shown. Unpaired *t*-test with Welch's correction. Error bars are s.d. Experiments were performed at 20˚C. Source data are provided in S1 Table.

highlighting the conserved role of FOXO in safeguarding oocyte health. In this study, depletion of *cyd-1* in the *daf-2* worms (low IIS) resulted in a DAF-16-dependent sterility due to the arrest of germ cells at the pachytene stage of meiosis-I. The canonical components of the IIS/PI3K pathway, with a predominant role of DAF-16a isoform, were found to be required to mediate this germline arrest. Noteworthily, the oocytes of *daf-16;daf-2* mutants that escape the pachytene arrest induced by *cyd-1* KD exhibit poor quality, highlighting the importance of DAF-16 in sensing perturbations in the CYD-1 levels and halting oogenesis, thereby preventing the production of unhealthy progenies (**Fig 7**). The role of FOXO proteins has been well documented as a checkpoint, where in various cell line models it orchestrates cell-cycle arrest and repair in response to DNA damage and oxidative stress [108–111]. We have earlier shown that DAF-16 responds to somatic DNA damage and arrests germline to protect the progeny [40]. Also, on sensing the unavailability of food, newly hatched worms utilize DAF-16 to arrest at the L1 diapause stage and resume development upon re-feeding [112,113]. Similarly, DAF-16 has a crucial role in another well-studied alternative third larval diapause stage; dauer, that forms in response to high population density or low food availability [114]. In both the above instances, crowding and low food availability would potentially jeopardize the success of the next generation. Thus, these instances reinforce the role of FOXO/DAF-16 as a custodian of both somatic as well as reproductive fitness, ultimately facilitating the efficient perpetuation of the genome.

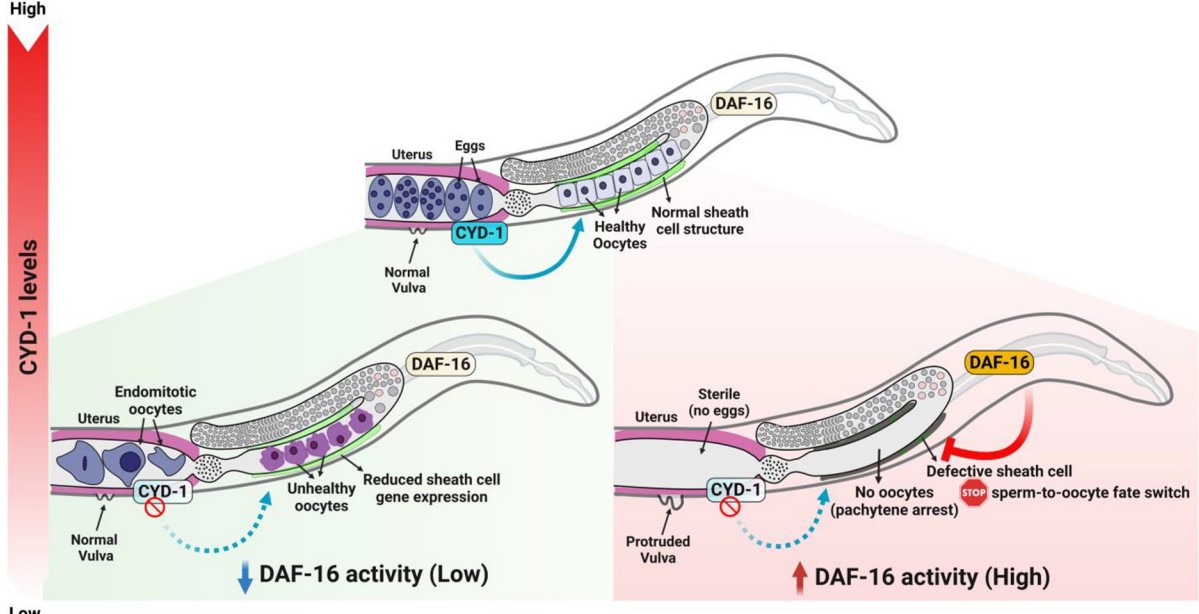

**Fig 7. Summary model.** Model showing combined effect of CYD-1 levels and DAF-16 activation state on the oogenesis and oocyte quality. Under wild-type conditions (normal insulin signaling) CYD-1 is required to maintain the oocyte health. Depletion of *cyd-1* leads to a compromise in oocyte quality and an increase in endomitotic oocytes in the uterus. Interestingly, when DAF-16 is activated (in *daf-2* worms), lowering CYD-1 levels results in DAF-16-dependent sheath cell defect which may possibly cause a failure in the sperm-to-oocyte fate switch, thereby halting oogenesis (pachytene arrest). Schematics in Fig 7 were created using BioRender.com.

Inter-tissue communication ensures an effective integration of environmental cues for optimal reproductive decisions. For example, activation of DAF-16 is required in the muscle and intestine for oocyte quality maintenance [18] and in the uterus for maintenance of germline stem cell (GSC) pool with age [22]. In line with this, we also find that CYD-1 regulates oocyte quality and pachytene arrest of the germ cells, cell non-autonomously from the somatic gonad (uterus) under wild-type and low IIS, respectively (**Fig 7**). We have previously demonstrated that somatic DNA damage triggers DAF-16/FOXO-dependent stalling of oogenesis under low IIS in a cell non-autonomous manner [40]. In a different context, somatic ERK/MPK-1 [115] and KRI-1 [116] have been shown to influence germ cell proliferation and death, respectively. The soma-germ cell communication is also vital for mammalian reproductive competence where the oocyte and surrounding somatic granulosa cells (GC) and the cumulus cells (CC) crosstalk to coordinate the growth and development of the follicle [117–120].

Intriguingly, contrary to the widely appreciated pro-longevity role of DAF-16, we observed activated DAF-16-dependent disruption of the somatic gonad morphology and gene expression upon *cyd-1* depletion under low IIS. We noticed abnormalities in the vulva (protruding vulva/vulvaless) and defects in the surrounding gonadal sheath cell structure. Consistent with this, we found transcriptional down-regulation of many genes important for somatic gonad development upon *cyd-1* depletion in *daf-2* worms, in a DAF-16-dependent manner (**Fig 7**). Activated DAF-16 led to complete abrogation of the hermaphrodite sheath cell-specific *lim-7p*::GFP and spermatheca-specific FKH-6::GFP expression in the *cyd-1*-depleted *daf-2* worms; both the structural defects in the gonadal sheath cells as well as gene expression alterations were reversed in the *cyd-1*-depleted *daf-16;daf-2* worms. Consistent with the role of sheath cells in the sperm-to-oocyte fate switch, we observed the downregulation of genes crucial for the sperm-to-oocyte switch in *cyd-1*-depleted *daf-2* worms. Notably, the removal of DAF-16 in these worms restored the transcript levels of these genes to normal levels, possibly enabling the pachytene exit of oogenic germ cells (**Fig 7**). At first, this appears strikingly counter-intuitive to the traditionally recognized role of DAF-16 in the maintenance of somatic and reproductive health. However, DAF-16 is well-established as a sentinel, often arresting the cells or halting development on encountering stresses such as starvation, DNA damage or oxidative stress, only to provide time for damage repair or resume growth/development upon return of favorable environmental conditions. Therefore, we speculate that activated DAF-16 may sense the perturbations in the levels of cell cycle proteins (such as CYD-1) and lead to exacerbation of defects in the somatic gonad, potentially driving a failure in the sperm-to-oocyte fate switch as a means of halting the production of poor-quality oocytes, thereby ensuring progeny fitness. Defects in cell cycle proteins in the somatic tissue may be interpreted as a potential threat to the progeny as such deficiencies are unlikely to be rectified.

In recent years, more women are postponing childbearing to later ages, bringing the problems of reproductive aging into focus. Also, there is a surge in cases of ovarian disorders like Polycystic Ovarian Syndrome (PCOS) [121] that affect oocyte quality [122]. It therefore becomes imperative to study pathways involved in oocyte quality maintenance to delay reproductive aging. The current study emphasizes the crucial role of the interplay between cell-cycle proteins and nutrient signaling in shaping reproductive outcomes. Our findings uncover a previously unidentified role of somatic CYD-1 in regulating reproductive fitness. Additionally, upon sensing cell cycle perturbations, activated DAF-16-dependent somatic gonadal tissue deterioration, may prevent the formation of low-quality oocytes. From an evolutionary standpoint, such mechanisms could be a strategy for the elimination of faulty progeny ensuring survival and propagation of healthy offspring in a species. In the future, it will be compelling to explore the mechanism through which cyclin D/CYD-1 and FOXO/DAF-16 impact such soma-germline signaling.

## Materials and methods

### *C. elegans* strain maintenance

Unless otherwise mentioned, all the *C. elegans* strains were maintained and propagated at 20˚C on *E. coli* OP50 using standard procedures (Stiernagle, 2006). *C. elegans* strains were obtained from Caenorhabditis Genetics Center (CGC) (Table 1) or generated in-house using standard cross over techniques (Table 2).

### Preparation of RNAi plates

RNAi plates were poured using an autoclaved nematode growth medium supplemented with 100 μg/ml ampicillin and 2 mM IPTG. Plates were allowed to dry at room temperature for 1 day. Bacterial culture harboring a particular RNAi construct was grown in Luria Bertani (LB) media supplemented with 100 μg/ml ampicillin and 12.5 μg/ml tetracycline, overnight at 37˚C

**Table 1. The list of strains obtained from Caenorhabditis Genetics Center (CGC) for use in the study.**

| S. No. | CGC Id | Genotype | Source |
|---|---|---|---|
| 1 | N2 var. Bristol | wild-type | CGC |
| 2 | CB1370 | *daf-2(e1370) III* | CGC |
| 3 | HT1890 | *daf-16 (mgDf50) I; daf-2 (e1370) III* | CGC |
| 4 | CU1546 | *smIs34 [ced-1p::ced-1::GFP + rol-6(su1006)]* | CGC |
| 5 | JT9609 | *pdk-1(sa680) X* | CGC |
| 6 | TJ1052 | *age-1(hx546) II* | CGC |
| 7 | DCL569 | *mkcSi13 [sun-1p::rde-1::sun-1 3'UTR + unc-119(+)] II; rde-1(mkc36) V* | CGC |
| 8 | NL2099 | *rrf-3(pk1426) II.* | CGC |
| 9 | NK640 | *rrf-3(pk1426) II; unc-119(ed4) III; rde-1(ne219) V; qyIs102[fos-1ap::rde-1(genomic) + myo-2::YFP + unc-119(+)]* | CGC |
| 10 | NL3511 | *ppw-1(pk1425) I* | CGC |
| 11 | OH441 | *otIs45 [unc-119::GFP]* | CGC |
| 12 | NR350 | *rde-1(ne219) V; kzIs20 [hlh-1p::rde-1 + sur-5p::NLS::GFP]* | CGC |
| 13 | NR222 | *rde-1(ne219) V; kzIs9 [(pKK1260) lin-26p::NLS::GFP + (pKK1253) lin-26p::rde-1 + rol-6(su1006)]* | CGC |
| 14 | VP303 | *rde-1(ne219) V; kbIs7 [nhx-2p::rde-1 + rol-6(su1006)]* | CGC |
| 15 | TU3401 | *sid-1(pk3321) V; uIs69 [pCFJ90 (myo-2p::mCherry) + unc-119p::sid-1] V* | CGC |
| 16 | WM27 | *rde-1(ne219) V* | CGC |
| 17 | SV314 | *rol-1(e91) cyd-1(he112)/mnC1 [dpy-10(e28) unc-52(e444)] II* | CGC |
| 18 | DZ325 | *ezIs2 [fkh-6::GFP + unc-119(+)) III; him-8(e1489) IV* | CGC |
| 19 | DZ224 | *him-8(e1489) IV; ezIs1 [K09C8.2::GFP + rol-6(su1006)] X* | CGC |
| 20 | PS3352 | *syIs50 [cdh-3::GFP + dpy-20(+)]* | CGC |
| 21 | DG1575 | *tnIs6 [lim-7::GFP + rol-6(su1006)]* | CGC |
| 22 | JU2039 | *mfIs70 [lin-31p::rde-1 + myo2p::GFP] IV; rde-1(ne219) V* | CGC |
| 23 | JK4143 | *qIs57 [lag-2p::GFP] II; rde-1(ne219) V; qIs140 [lag-2p::rde-1 + rol-6(su1006)]* | CGC |
| 24 | SU93 | *jcIs1 [ajm-1::GFP + unc-29(+) + rol-6(su1006)] IV* | CGC |
| 25 | HT1881 | *daf-16(mgDf50) I; daf-2(e1370); unc-119(ed3) III; lpIs12 [daf-16a::RFP + unc-119(+)]* | CGC |
| 26 | HT1882 | *daf-16(mgDf50) I; daf-2(e1370); unc-119(ed3) III; lpIs13 [daf-16b::CFP + unc-119(+)]* | CGC |
| 27 | HT1883 | *daf-16(mgDf50) I; daf-2(e1370); unc-119(ed3) III; lpIs14 [daf-16f::GFP + unc-119(+)]* | CGC |
| 28 | CB4108 | *fog-2(q71) V* | CGC |
| 29 | NK741 | *rrf-3(pk1426) II; unc-119(ed4) III; rde-1(ne219) V; qyIs138 [unc-62p::rde-1(genomic) + myo-2::YFP + unc-119(+)]* | CGC |

**Table 2. The strains listed below were generated in-house using standard cross-over techniques.**

| S. No. | Genotype |
|---|---|
| 1 | *daf-2(e1370) III; WM27: rde-1(ne219) V* |
| 2 | *daf-2(e1370) III; DCL569: mkcSi13 [sun-1p::rde-1::sun-1 3'UTR + unc-119(+)] II; rde-1(mkc36) V* |
| 3 | *daf-2(e1370)III; NR350: rde-1(ne219) V; kzIs20 [hlh-1p::rde-1 + sur-5p::NLS::GFP]* |
| 4 | *daf-2(e1370) III; NR222: rde-1(ne219) V; kzIs9 [(pKK1260) lin-26p::NLS::GFP + (pKK1253) lin-26p::rde-1 + rol-6(su1006)]* |
| 5 | *daf-2(e1370) III; VP303: rde-1(ne219) V; kbIs7 [nhx-2p::rde-1 + rol-6(su1006)]* |
| 6 | *daf-2(e1370) III; TU3401: sid-1(pk3321) V; uIs69 V [pCFJ90 (myo-2p::cherry) + unc-119p::sid-1]* |
| 7 | *daf-2(e1370) III; NK640: rrf-3(pk1426) II; unc-119(ed4) III;rde-1(ne219) V;qyIs102 [fos-1ap::rde-1(genomic) + myo-2::YFP + unc-119(+)]* |
| 8 | *daf-2(e1370) III; JU2039: mfIs70 [lin-31p::rde-1 + myo2p::GFP] IV; rde-1(ne219) V.* |
| 9 | *daf-2(e1370) III; JK4143: qIs57 [lag-2p::GFP] II; rde-1(ne219) V; qIs140 [lag-2p::rde-1 + rol-6(su1006)]* |
| 10 | *daf-2(e1370) III; CU1546: smIs34 [ced-1p::ced-1::GFP + rol-6(su1006)]* |
| 11 | *daf-2(e1370) III; DZ325: ezIs2 [fkh-6::GFP + unc-119(+)] III; him-8(e1489) IV* |
| 12 | *daf-2(e1370) III; DZ224: him-8(e1489) IV; ezIs1 [K09C8.2::GFP + rol-6(su1006)] X* |
| 13 | *daf-2(e1370) III; PS3352: syIs50 [cdh-3::GFP + dpy-20(+)]* |
| 14 | *daf-2(e1370) III; DG1575: tnIs6 [lim-7::GFP + rol-6(su1006)]* |
| 15 | *daf-2(e1370) III; SU93: jcIs1 [ajm-1::GFP + unc-29(+) + rol-6(su1006)] IV* |
| 16 | *daf-16(mgDf50) I; daf-2(e1370) III; CU1546: smIs34 [ced-1p::ced-1::GFP+ rol-6(su1006)]* |
| 17 | *daf-16(mgDf50) I; daf-2(e1370) III; DZ325: ezIs2 [fkh-6::GFP + unc-119(+)] III; him-8(e1489) IV* |
| 18 | *daf-16(mgDf50) I; daf-2(e1370) III; PS3352: syIs50 [cdh-3::GFP + dpy-20(+)]* |
| 19 | *daf-16(mgDf50) I; daf-2(e1370) III; DG1575: tnIs6 [lim-7::GFP + rol-6(su1006)]* |
| 20 | *daf-16(mgDf50) I; daf-2(e1370) III; SU93: jcIs1 [ajm-1::GFP + unc-29(+) + rol-6(su1006)] IV* |
| 21 | *daf-2(e1370) III; NL2099: rrf-3(pk1426) II* |
| 22 | *daf-2(e1370) III; NK741: rrf-3(pk1426) II; unc-119(ed4) III; rde-1(ne219) V; qyIs138 [unc-62p::rde-1 (genomic) + myo-2::YFP + unc-119(+)]* |

in a shaker incubator. Saturated primary cultures were re-inoculated the next day ($1/50^{th}$ volume) in fresh LB media containing 100 μg/ml ampicillin and grown in a 37°C shaker until $OD_{600}$ reached 0.5–0.6. The bacterial cells were pelleted down by centrifuging the culture at 3214 g for 5 minutes at 4°C and resuspended in $1/10^{th}$ volume of M9 buffer containing 100 μg/ml ampicillin and 1 mM IPTG. Around 300 μl of this suspension was seeded onto RNAi plates and left at room temperature for 2 days for drying, followed by storage at 4°C till further use.

## Hypochlorite treatment to obtain synchronize worm population

Gravid adult worms, initially grown on *E. coli* OP50 bacteria, were collected using M9 buffer in a 15 ml falcon tube. Worms were washed twice by first centrifuging at 652 g for 60 seconds followed by resuspension of the worm pellet in 1X M9 buffer. After the second wash, the worm pellet was resuspended in 3.5 ml of 1X M9 buffer and 0.5 ml 5N NaOH and 1 ml of 4% Sodium hypochlorite solution was added. The mixture was vortexed for 4–5 minutes until the entire worm body dissolved, leaving behind the eggs. The eggs were washed 5–6 times, by first centrifuging at 1258 g, the 1X M9 was aspirated out, followed by resuspension in fresh 1X M9 buffer to remove traces of bleach and alkali. After the final wash, eggs were kept in 15 ml falcons with ~ 10 ml of 1X M9 buffer and kept on rotation ~15 r.p.m for 17 hours to obtain L1 synchronized worms for all strains. The L1 worms were obtained by centrifugation at 805 g followed by resuspension in approximately 200–300 μl of M9 and added to respective RNAi plates.

## RNA isolation

Worms grown on respective RNAi were collected using 1X M9 buffer and washed thrice to remove bacteria. Trizol reagent (200 μl; Takara Bio, Kusatsu, Shiga, Japan) was added to the 50 μl worm pellet and subjected to three freeze-thaw cycles in liquid nitrogen with intermittent vortexing for 1 minute to break open the worm bodies. The samples were then frozen in liquid nitrogen and stored at -80°C till further use. Later, 200 μl of Trizol was again added to the worm pellet and the sample was vigorously vortexed for 1 minute. To this, 200 μl of chloroform was added and the tube was gently inverted several times followed by 5 minutes of incubation at room temperature. The sample was then centrifuged at 12000 g for 15 minutes at 4°C. The RNA containing the upper aqueous phase was gently removed into a fresh tube without disturbing the bottom layer and interphase. To this aqueous solution, an equal volume of isopropanol was added and the reaction was allowed to sit for 10 minutes at room temperature followed by centrifugation at 12000 g for 15 minutes at 4°C. The supernatant was carefully discarded without disturbing the RNA-containing pellet. The pellet was washed using 1 ml 70% ethanol solution followed by centrifugation at 8000g for 10 minutes at 4°C. The RNA pellet was further dried at room temperature and later dissolved in autoclaved MilliQ water followed by incubation at 65°C for 10 minutes with intermittent tapping. The concentration of RNA was determined by measuring absorbance at 260 nm using a NanoDrop UV spectrophotometer (Thermo Scientific, Waltham, USA) and the quality was checked using denaturing formaldehyde-agarose gel.

## Gene expression analysis using quantitative real-time PCR (QRT-PCR)

First-strand cDNA synthesis was carried out using the Iscript cDNA synthesis kit (Biorad, Hercules, USA) following the manufacturer's guidelines. The prepared cDNA was stored at -20°C. Gene expression levels were determined using the Brilliant III Ultra-Fast SYBR Green QPCR master mix (Agilent, Santa Clara, USA) and Agilent AriaMx Real-Time PCR system (Agilent, Santa Clara, USA), according to manufacturer's guidelines. The relative expression of each gene was determined by normalizing the data to *actin* expression levels. The list of primers is summarized in S4 Table.

## Measurement of cell corpses using CED-1::GFP

The analysis of engulfed cell corpses was performed by utilizing transgenic worms expressing CED-1::GFP, a transmembrane protein found on surrounding sheath cells that are responsible for engulfing cell corpses. *ced-1p::ced-1::GFP(smIs34)* worms were bleached and their eggs were allowed to hatch in 1X M9 buffer for 17 hours to obtain L1 synchronized worms. Approximately 100 L1 worms were placed onto control or *cyd-1* RNAi in triplicates. On the day-1 of adulthood, worms were imaged in Z-stack using 488 nm laser to excite GFP on LSM980 confocal microscope (Carl Zeiss, Oberkochen, Germany). The images were converted into a maximum intensity projection (MIP). The number of cell corpses per gonadal arm was quantified manually.

## DAPI staining

Worms were cultured on specific RNAi plates from the L1 stage onward. On day 1 of adulthood, worms were collected in a 1.5 ml Eppendorf tube with 1X M9 buffer and allowed to settle. Using a glass Pasteur pipette, the 1X M9 buffer was carefully removed, leaving approximately 100 μl of worm suspension. Subsequently, 1 ml of ice-cold 100% methanol was added to the worm pellet, and the mixture was incubated at -20°C for 30 minutes. The

methanol-fixed worm pellet was then placed on a glass slide, and after the methanol had evaporated, Fluoroshield with DAPI (from Invitrogen, Carlsbad, USA) was applied for staining. A coverslip was carefully placed on top (avoiding bubbles) and sealed with transparent nail paint.

To stain dissected gonads, worms were positioned on a glass slide in 1X M9, and the gonads were obtained by carefully cutting the pharynx or tail end of the worm using a sharp 25G needle. After this 500 μl of chilled 100% methanol was added to the slide and allowed to evaporate. Subsequently, Fluoroshield with DAPI (Invitrogen, Carlsbad, USA) was added for staining and a coverslip was carefully placed on top (avoiding bubbles) and sealed with transparent nail paint. Finally, the images were captured using a 405 nm laser excitation DAPI on LSM980 confocal microscope (Carl Zeiss in Oberkochen, Germany).

## Brood size, reproductive span, and egg hatching

**Brood size.** Worms were grown on control or *cyd-1* RNAi from L1 onwards and upon reaching the young adult stage, five worms were picked onto fresh RNAi plates, in triplicates, and allowed to lay eggs for 24 hours. The worms were then transferred to fresh plates every day until worms ceased to lay eggs, and the eggs laid on the previous day's plate were counted. These plates were again counted after 48 hours to document the number of hatched worms. The total number of hatched progenies per worm is defined as brood size.

## Reproductive span

The number of progenies per worm was plotted for each day to monitor the reproductive span. For the mated reproductive span, the hermaphrodites were mated on 35 mm cross plates with fresh males (males were grown on control RNAi) (1:3 ratio of hermaphrodite:male) each day for 4 h and shifted back to 60 mm RNAi plates. The rest was performed according to the reproductive span of selfed worms. The total number of days when worms were capable of producing viable offspring was recorded as their reproductive span.

## Egg hatching

The quality of eggs was assessed by determining the percentage of hatched eggs. Almost 50 adult worms per condition were allowed to lay eggs for an hour. The mother worms were then sacrificed and the number of eggs on respective plates were counted. After 48 hours the number of hatched progenies were counted, and the percentage of hatched eggs were calculated for each condition.

## Analysis of oocyte morphology

Worms were grown from L1 onwards on control or *cyd-1* RNAi. Microscopic images of oocytes on the third day of adulthood (except for S3B and S3C Fig where day-1 oocytes were imaged) were captured using differential interference contrast (DIC) settings (Carl Zeiss, Oberkochen, Germany). These oocytes were classified into three groups based on various characteristics such as cavities, shape, size, and organization. Depending on the severity of the observed features, oocytes were classified as either normal, mild, or severe. Oocytes without cavities, small size, or disorganization were labeled as normal. Oocytes exhibiting the presence of either two small oocytes, two cavities, or two disorganized oocytes were categorized as having a mild phenotype. Those with more than two cavities, small oocytes, or misshapen oocytes were considered to have a severe phenotype.

## Quantification of fertile worms

Approximately 100 synchronized L1 worms were placed onto different RNAi plates, in triplicates. On the day-1 adult stage, bright-field images were captured (Carl Zeiss, Oberkochen, Germany). Worms carrying over five eggs in their uterus were categorized as fertile.

## Germ cell count

Around 100 L1-staged worms were poured onto control or *cyd-1* RNAi plates in triplicate. Utilizing a 405 nm laser to excite DAPI in a confocal microscope (Carl Zeiss, Oberkochen, Germany), images of DAPI stained day-1 adult germline were acquired. Z stacked images (0.13 μM sections between the starting plane and end plane) of germline were obtained followed by Maximum Intensity projection (MIP) of each z-stack in the XY plane to create a superimposed image. Quantification of germ cells per gonadal arm was done using Fiji (National Institute of Health) in different regions based on their morphology, namely the Mitotic Zone (spanning from the distal tip to the transition zone), the Transition zone (extending from the beginning to the end of crescent-shaped nuclei), and the pachytene zone (from the end of the transition zone to the turn region).

## Sperm count

DAPI stained early day-1, day-2 or day-3 adult gonads were imaged using 405 nm laser to excite DAPI on a confocal microscope (Carl Zeiss, Oberkochen, Germany). Z stacked images (0.13 μM sections between the starting plane and end plane) of germline were obtained followed by Maximum Intensity projection (MIP) of each z-stack in the XY plane to create a superimposed image. Sperms per gonadal arm were quantified using Fiji (National Institute of Health).

## Scoring of Endomitotic oocyte (emo)

Approximately 50 L1-staged worms were placed onto control or *cyd-1* RNAi plates in triplicate. Day-3 adult worm gonads (except for S3D and S3E Fig where day-1 worm gonads were imaged) were imaged using differential interference contrast (DIC) settings in a microscope (Carl Zeiss, Oberkochen, Germany). Endomitotic oocytes in the uterus were identified by following features: a lack of distinct nucleus, nuclei may appear disorganized, lobulated, or enlarged [56]. Alternatively, day-3 adults were DAPI stained and subsequently imaged in a Z-stack using 405 nm laser to excite DAPI on an LSM980 confocal microscope (Carl Zeiss, Oberkochen, Germany). Nuclei that appeared enlarged and exhibited dense DAPI staining [123], were identified as endomitotic oocytes (emo). For the purpose of scoring, the images were converted into a maximum intensity projection (MIP).

## *unc-119*::GFP positive emo

Approximately 50 L1-staged *unc-119*::*gfp* (neuronal fate marker) transgenic worms were placed onto control or *cyd-1* RNAi plates in triplicates. Day-3 adult worm gonads were imaged using 488 nm laser to excite GFP as well as differential interference contrast (DIC) settings in a fluorescence microscope (Carl Zeiss, Oberkochen, Germany). Endomitotic oocytes in the uterus were identified by following features: a lack of distinct nucleus, nuclei may appear disorganized, lobulated or enlarged [56]. The number of worms having endomitotic oocytes as well as that expressed *unc-119*::GFP in the emo were counted. Percentage of worms expressing GFP in emo was calculated for each condition.

## Imaging of transgenic reporter strains

Synchronized L1 populations of transgenic reporter strains were grown on respective RNAi plates. For imaging, the worms were mounted on agar pads with 20 mM sodium azide and a coverslip was placed on the paralyzed worms. The coverslip was secured with a transparent nail paint. All images were acquired in a Z-stack using 488 nm laser to excite GFP on LSM980 confocal microscope (Carl Zeiss, Oberkochen, Germany). The images were converted into a maximum intensity projection (MIP). To qualitatively quantify *lim-7p*::GFP (sheath cell marker) and FKH-6::GFP (hermaphrodite spermatheca) expression, the day-1 or day-3 adult gonads were scored as having normal expression, reduced expression (less intensity than normal) or missing (no GFP expression). In the case of *cdh-3*::GFP (a marker for vulval cells), we used late-L3 staged worms and scored their gonads based on whether they displayed normal expression, abnormal expression (distorted pattern), or missing (no GFP expression). Similar categorization was utilized to assess the AJM-1::GFP (adherens junction marker) expression in early day-1 adult gonads.

## Phalloidin staining and sheath cell structure

Worms were grown on control and *cyd-1* RNAi L1 onwards. Day-1 or day-3 adult worms were collected in a 1.5 ml Eppendorf tube with 1X M9 buffer and allowed to settle. Using a glass Pasteur pipette, excess 1X M9 buffer was carefully removed, and worms were transferred to a glass slide. Using a 25G needle, worms were excised at the pharynx or near tail region, causing gonad arms to extrude from the body. The dissected gonad arms were transferred to 5 ml glass tubes and fixed in 4% formaldehyde for 20 minutes at room temperature and then washed thrice with 0.1% 1X PBST. After washing, fixed gonad arms were transferred to 1 ml glass tubes and stained with Oregon Green 488 Phalloidin (Molecular Probes, Invitrogen, Life Technologies, Grand Island, NY, USA) at a final concentration of 0.4 units/ml and placed in the dark at 4°C overnight followed by 30 minutes incubation at RT. Gonad arms were then washed twice with PBST, and extra liquid was removed and gonad arms were transferred onto slides with a drop of Fluoroshield with DAPI (Invitrogen, Carlsbad, USA). Coverslip was placed on the mounted worms and sealed using nail-paint. Phalloidin marks the actin structure. To analyze sheath cell structure gonads were imaged in a Z-stack using 488 nm laser to excite GFP on LSM980 confocal microscope (Carl Zeiss, Oberkochen, Germany). The images were converted into a maximum intensity projection (MIP). Sheath cell structure was scored as normal or mild defect (slight clustering or loosening of actin filaments) or severe defect (highly clustered or loosened actin filaments). To examine vulval morphology, the images were captured with a focus on the vulva-uterine region.

## Vulva morphology

Worms were grown from L1 onwards on control or *cyd-1* RNAi. Microscopic images of vulva on the early day-1 of adulthood were captured using differential interference contrast (DIC) settings on a microscope (Carl Zeiss, Oberkochen, Germany). These vulvas were classified as normal or abnormal (protruding vulva/vulvaless) based on its structure and organization.

## RNA sequencing

Synchronized late-L4 worms grown on control or *cyd-1* RNAi were collected using 1X M9 buffer, after washing it thrice to remove bacteria. Total RNA was isolated from these worm pellets using the Trizol method. The concentration of RNA was determined by measuring absorbance at 260 nm using a NanoDrop UV spectrophotometer (Thermo Scientific, Waltham,

USA) and RNA quality was checked using RNA 6000 NanoAssay chip on a Bioanalyzer 2100 machine (Agilent Technologies, Santa Clara, USA). RNA above RNA integrity number = 9 was included in the study. A 1.5 μg aliquot of RNA samples was used for polyA mRNA isolation from total RNA using NEBNext Poly(A) mRNA Magnetic Isolation Module (Catalog no-E7490L) according to the maufacturer's protocol. The NEBNext Ultra II Directional RNA Library Prep kit (Catalog no-E7760L) was used to construct libraries according to the manufacturer's instructions. NovaSeq6000 NGS platform (Illumina Inc., San Diego, California, USA) with 2X150 paired end chemistry was used to generate 30 million paired end reads corresponding to 9 Gb data per sample. The RNA sequencing data are available at the https://www.ebi.ac.uk/biostudies/arrayexpress with accession number E-MTAB-13172.

## RNA-seq Analysis

Sequencing reads were subjected to quality control using the Fastp kit [124]. The quality of reads was assessed and visualized by FASTQC (https://www.bioinformatics.babraham.ac.uk/projects/fastqc). Adapter sequence was removed by Trimmomatic tool [125]. For each of our samples, reads were aligned to the WBcel235 cell genome using STAR-2.7.10b [126] with an average 94.4% alignment rate. Gene counts were obtained with the quantMode GeneCounts option. Differential Gene expression analysis was performed using DeSeq2 [127] package in R. Differentially expressed genes were defined as those with padj values below 0.06. Genes with a cut-off of $\log_2$(fold change) $> 0.6$ and $\log_2$(fold change) $< -0.6$ were considered as upregulated and downregulated genes, respectively. For downstream analysis, the function variance stabilizing transformations (VST) in the DeSeq2 package was implemented. Wormbase GeneIDs were mapped to their gene symbols by AnnotationaDbi and *C. elegans* database (org.Ce.eg.db) packages in R. Counts function was used to normalize the gene counts. Gene Set Enrichment analysis was performed using gseGO function of clusterProfiler v4.6.2 [128] in R. Volcano plots and dot plots were plotted with ggplot2 (https://ggplot2.tidyverse.org) in R. For volcano plots, differentially expressed genes (padj $<0.1$ and $|\log_2$(fold change)$|>0.6$) were indicated in green and purple colors. Dot plots were made with Enriched GO terms on y-axis and Normalized Enrichment Score (NES) on x-axis and size and color as padj value.

## Statistical tests

All statistical tests were performed utilizing the built-in functions of GraphPad Prism 10.1.0. In instances where we compared two conditions (continuous data), we applied a Student's *t*-test with Welch's correction, making no assumptions about consistent standard deviations. When comparing categorical data, we used the chi-square test. When comparing multiple conditions (continuous), we employed a Two-way ANOVA with Tukey's multiple comparison test.

## Supporting information

**S1 Fig. Role of CYD-1 in maintenance of oocyte quality. (A)** RNAi screen for cyclins that may interact with the IIS. Twelve cyclin genes were knocked down in WT, *daf-16(mgdf50)*, *daf-2(e1370)* and *daf-16(mgdf50);daf-2(e1370)* using RNAi. The tick mark indicates fertile worms while the cross mark indicates sterility when grown on respective RNAi from L1 onwards. **(B)** Representative fluorescent images of DAPI-stained germ line of *daf-2(e1370)* and *daf-16(mgdf50);daf-2(e1370)* worms grown on control or *cye-1* RNAi. Oocytes are boxed for clarity. White arrows point towards oocytes while yellow shows the absence of oocytes. Sp denotes sperms. **(C,D)** Representative images of DAPI-stained dissected gonadal arm of WT worms (day-1 adult) grown on control or *cyd-1* RNAi. Mitotic zone (MT), transition Zone

(TZ), and pachytene zone (PZ) germ cells are marked with a solid line and white dashed line mark oocytes (C). Quantification of germ cells in each zone (D). Each point represents the number of germ cells in the respective zones (n = 17). Unpaired *t*-test with Welch's correction. **(E,F)** Representative images showing apoptotic cells (marked by arrows) in the gonadal arm of *ced-1*::*gfp* worms (day-1 adult) grown on control or *cyd-1* RNAi (E). Quantification for apoptotic cells per gonadal arm (F) The average of three biological replicates is shown (n ≥ 15 for each replicate). Unpaired *t*-test with Welch's correction. **(G)** Oocyte quality score based on morphology. The quality was categorized as normal, mild or severe based on the presence of cavities, shape and organization of oocytes. Normal = No cavities/not misshapen/not disorganized, mild = either with cavities/ are misshapen/disorganized (≤ 2 instances per worm), severe = either with cavities/ are misshapen/disorganized (≥ 3 instances per worm). **(H)** Representative DAPI-stained gonads of WT hermaphrodites (day-3 adult) and *fog-2(q71)* females grown on control or *cyd-1* RNAi at 25°C. Oocytes are boxed for clarity. White arrows point towards oocytes while yellow arrows point towards endomitotic oocytes (emos). **(I)** Oocyte quality scores (based on morphology as shown in S1G Fig) for WT (day-3) worms grown on control RNA or *cyd-1* RNAi from L1 onwards or transferred from control to *cyd-1* RNAi at Late-L4 larval stage. Combined data from three biological repeats (n ≥ 36) is plotted. Chi-square analysis was used to compare between groups. **(J)** Quantification of endomitotic oocytes for WT (day-3) worms grown on control RNA or *cyd-1* RNAi from L1 onwards or transferred from control to *cyd-1* RNAi at Late-L4 larval stage. Combined data from two biological replicates (n ≥ 36) is plotted. Chi-square analysis was used to compare between groups. Scale bars:20 μm. Error bars are s.d. Experiments were performed at 20°C except S1H which was performed at 25°C. Source data are provided in S1 Table.
(TIF)

**S2 Fig. Tissue-specific role of CYD-1 in germline quality assurance. (A)** The total number of hatched progenies in *rde-1(mkc36);sun-1p*::*rde-1* worms (germline-specific RNAi) grown on control or *cyd-1* RNAi. Average of three biological repeats. Unpaired *t*-test with Welch's correction. **(B)** The total number of hatched progenies in *ppw-1(pk1425)* worms (soma-specific RNAi) grown on control or *cyd-1* RNAi. Average of three biological repeats. Unpaired *t*-test with Welch's correction. **(C)** Representative images of *ppw-1(pk1425)* (soma-specific RNAi), or *rde-1(mkc36);sun-1p*::*rde-1* (germline-specific RNAi) worms (day-1 adult) grown on control, *egg-5* or *dpy-7* RNAi. KD of *egg-5* led to non-viable progeny in the germline-specific RNAi strain but not in the soma-specific RNAi strain, while KD of *dpy-7* led to dumpy phenotype in the soma-specific RNAi strain but not in the germline-specific RNAi strain. Scale bar 100 μm. **(D,E)** DIC images showing oocyte morphology of *rrf-3(pk1426);rde-1 (ne219);nhx-2p*::*rde-1* (intestine-specific RNAi) or *rrf-3(pk1426);rde-1(ne219);hlh-1p*::*rde-1* (muscle-specific RNAi) worms (day-3 adult) grown on control or *cyd-1* RNAi. White arrows mark oocytes. Scale bar 20 μm (D). Oocyte quality score (based on morphology as shown in S1G Fig) (E). Combined data from three biological replicates (n ≥ 34) is plotted. Chi-square analysis was used to compare between groups. **(F,G)** DIC images showing the presence of unfertilized oocytes/endomitotic oocytes (emos) in the uterus of *rrf-3(pk1426);rde-1(ne219); fos-1ap*::*rde-1(genomic)* (uterine tissue-specific RNAi) worms (day-3 adult) grown on control or *cyd-1* RNAi. White arrows mark unfertilized oocytes in the uterus while yellow arrows mark endomitotic oocytes (emos). Scale bar 20 μm (F). Quantification for endomitotic oocytes (G) Combined data from three biological replicates (n ≥ 24) is plotted. Chi-square analysis was used to compare between groups. **(H)** The total number of hatched progenies in *rrf-3 (pk1426);rde-1(ne219);fos-1ap*::*rde-1(genomic)* (uterine tissue-specific RNAi) worms grown on control or *cyd-1* RNAi. Average of four biological repeats. Unpaired *t*-test with Welch's

correction. **(I)** Representative images of *rrf-3(pk1426);rde-1(ne219);fos-1ap::rde-1* (uterine tissue-specific RNAi) worms (day-1 adult) grown on *egl-43*, *egg-5 or dpy-7* RNAi. KD of *egl-43* led to protruded vulva, while *egg-5* KD resulted in viable progeny and *dpy-7* KD led to non-dumpy progeny. Scale bar 100 μm. Error bars are s.d. Experiments were performed at 20˚C. Source data are provided in S1 Table.
(TIF)

**S3 Fig. Depletion of *cyd-1* leads to decreased oocyte quality and increased apoptosis without affecting the DNA damage response (DDR). (A)** RT-PCR analysis showing knockdown efficiency of *cyd-1* RNAi in *daf-2(e1370)* and *daf-16(mgdf50);daf-2(e1370)*. Expression levels were normalized to *actin*. Average of 5 biological replicates are shown. Two-way ANOVA-Tukey's multiple comparisons test. **(B,C)** DIC images showing oocyte morphology of *cyd-1 (he112)/+* (day-1 adult) worms grown on control or *cyd-1* RNAi. White arrows mark normal oocytes whereas yellow arrows mark oocytes with cavities or those that are misshapen or disorganized, indicative of poor quality (B). Oocyte quality score (based on morphology as shown in S1G Fig) (C). Combined data from three biological replicates (n ≥ 33) is plotted. Chi-square analysis was used to compare between groups. **(D,E)** DIC images showing eggs or endomitotic oocytes (emos) in *cyd-1(he112)/+* (day-1 adult) worms grown on control or *cyd-1* RNAi. White arrows mark normal eggs in the uterus whereas yellow arrows mark emos (D). Quantification for endomitotic oocytes. (E) Combined data from three biological replicates (n ≥ 20) is plotted. Chi-square analysis was performed to compare between groups. **(F)** Quantitative RT-PCR analysis for DNA damage response (DDR) genes in WT late-L4 staged worms grown on control or *cyd-1* RNAi. Expression levels were normalized to *actin*. Average of three biological replicates are shown. Unpaired *t*-test with Welch's correction. **(G)** The percentage of eggs hatched in control or *cyd-1* RNAi fed WT L4-staged worms exposed to different doses of IR (0, 20, 30, 40 Gy). The average of three biological replicates is shown ($n≥20$). Unpaired *t*-test with Welch's correction **(H,I)** Representative fluorescent and DIC merged images showing apoptotic cells (arrows) in the gonadal arm of *daf-2(e1370);ced-1*::*gfp and daf-16(mgdf50);daf-2 (e1370);ced-1*::*gfp* (day-1 adult) worms grown on control or *cyd-1* RNAi. Arrows mark apoptotic corpses. (H). Quantification for apoptotic corpses per gonadal arm (I). An average of three biological replicates are shown (n ≥ 17 for each replicate). Unpaired *t*-test with Welch's correction. Scale bars:20 μm. Error bars are s.d. Experiments were performed at 20˚C. Source data are provided in S1 Table.
(TIF)

**S4 Fig. Depletion of *cyd-1* leads to germline arrest upon lowering of canonical insulin signaling. (A,B)** DIC images showing oocyte morphology of *daf-16(mgdf50);daf-2(e1370)* (day-3 adult) worms grown on control or *cyd-1* RNAi. White arrows mark normal oocytes whereas yellow arrows mark oocytes with cavities or those that are misshapen or disorganized, indicative of poor quality (A). Oocyte quality score (based on morphology as shown in S1G Fig) (B). Combined data from three biological replicates (n ≥ 32) is plotted. Chi-square analysis was used to compare between groups. **(C,D)** DIC images showing eggs or endomitotic oocytes (emos) in *daf-16(mgdf50);daf-2(e1370)* (day-3 adult) worms grown on control or *cyd-1* RNAi. White arrows mark normal eggs in the uterus whereas yellow arrows mark emos (C). Quantification for endomitotic oocytes (D). Average of three biological replicates (n ≥ 20 for each replicate). Unpaired *t*-test with Welch's correction. **(E,F)** Representative fluorescent images of DAPI stained gonads of *age-1(hx546)* and *pdk-1(sa680)* (day-1 adult) worms grown on control or *cyd-1* RNAi. Oocytes are boxed for clarity. White arrows point towards oocytes while yellow shows the absence of oocytes. Sp denotes sperms (E). The percentage of fertile worms (F). Average of three biological replicates (n ≥ 30 for each replicate). Unpaired *t*-test with Welch's

correction. **(G)** Representative DAPI-stained gonads of *rrf-3(pk1426);daf-2(e1370)* worms (day-1 adult) grown on control or *cdk-4* RNAi. Oocytes are boxed for clarity. White arrows point towards oocytes while yellow shows the absence of oocytes. Scale bars:20 μm. Error bars are s.d. Experiments were performed at 20˚C. Source data are provided in S1 Table.
(TIF)

**S5 Fig. Depletion of *cyd-1* under low insulin signaling leads to defects in the egg-laying apparatus. (A)** The percentage of fertile worms upon tissue-specific *cyd-1* KD in *daf-2(e1370)*. The following promoters were used to drive the tissue-specific expression of *rde-1* (Vulva: *lin-31p*; DTC: *lag-2p*; Hypodermis: *lin-26p*; muscle: *hlh-1p*; intestine: *nhx-2p*) in a *rde-1(ne219)* mutant. For neuron-specific RNAi, *unc-119p* was used to drive the expression of *sid-1* cDNA in the neurons of *sid-1(pk3321)* mutant. Average of three biological replicates (n ≥ 25 per condition for each experiment). Unpaired *t*-test with Welch's correction. **(B,C)** DIC images of gonads of *daf-2(e1370)* and *daf-16(mgdf50);daf-2(e1370)* worms (day-1 adult) on control or *cyd-1* RNAi. White arrows point towards normal vulva while yellow points towards abnormal vulva (protruded). Scale bar 100 μm (B). Combined data from three biological replicates (n ≥ 22) is plotted. Chi-square analysis was used to compare between groups. **(D)** Representative fluorescent images of Phalloidin-stained (that marks the F-actin) gonads of *daf-2(e1370)* worms (day-1 adult) grown on control or *cyd-1* RNAi. White arrows show normal vulva muscle structure. Yellow arrows show defective vulva muscles. Scale bar 10 μm. **(E,F)** Representative fluorescent images of DAPI-stained gonads of *daf-2(e1370)* and *daf-16(mgdf50);daf-2(e1370)* (day-1 adult) worms grown on control or *egl-43* RNAi. Oocytes are boxed for clarity. White arrows point towards oocytes while yellow shows endomitotic oocytes. Sp denotes sperms. Scale bar 20 μm (E). The percentage of fertile worms (F). Average of three biological replicates (n ≥ 25 per condition for each experiment). Two-way ANOVA-Tukey's multiple comparisons test. Error bars are s.d. Experiments were performed at 20˚C. Source data are provided in S1 Table.
(TIF)

**S6 Fig. Lowering CYD-1 levels leads to DAF-16-dependent sheath cell defects in *daf-2*. (A-B)** Gene set enrichment analysis of differentially expressed genes in (A) *daf-2(e1370);rrf-3(pk1426);rde-1(ne219);unc-62p::rde-1(genomic)* or (B) *rrf-3(pk1426);rde-1(ne219);unc-62p::rde-1(genomic)* upon *cyd-1* KD as compared to control RNAi. **(C)** Volcano plot showing the magnitude [$\log_2$(FC)] and significance [$-\log_{10}$(P value)] of the genes that are differentially expressed in L4 stage of *rrf-3(pk1426);rde-1(ne219);unc-62p::rde-1(genomic)* worms, grown on control or *cyd-1* RNAi. **(D)** Sheath cell structure scoring scheme. Representative fluorescent images of Phalloidin-stained (that marks the F-actin) gonads of worms (day-1 adult). The quality was categorized as normal, mild or severe based on the structure and organization of actin filaments. Normal = No loosened or disorganized structure, mild = either loosened or disorganized actin filaments, severe = highly loosened or disorganized actin filaments. Yellow arrows point towards the defect. **(E,F)** Representative fluorescent and DIC merged images of gonads showing *lim-7p*::GFP (that marks the sheath cells) expression in *daf-2(e1370)* and *daf-16(mgdf50);daf-2(e1370)* worms (day-1 adult) grown on control and *sys-1* RNAi. The gonadal arm is outlined for clarity. Scale bar 20 μm (E). Quantification of the normal, reduced or missing expression of *lim-7p*::GFP (F). Combined data from three biological replicates (n ≥ 32) is plotted. Chi-square analysis was used to compare between groups. **(G,H)** Representative fluorescent and DIC merged images of gonads showing FKH-6::GFP (that marks the hermaphrodite spermatheca) expression in *daf-2(e1370)* and *daf-16(mgdf50);daf-2(e1370)* worms (day-1 adult) grown on control and *cyd-1* RNAi. Spermatheca is outlined for clarity. Scale bar 20 μm (G). Quantification of the normal or missing expression of FKH-6::GFP (H). Combined data

from three biological replicates (n ≥ 24) is plotted. Chi-square analysis was used to compare between groups. **(I,J)** Representative fluorescent images of Phalloidin-stained (that marks the F-actin) gonads of WT worms (day-3 adult) grown on control and *cyd-1* RNAi. Scale bar 20 μm (I). Quantification of the normal or defective sheath cell structure (as per scoring scheme in S6D Fig) (J) Combined data from three biological replicates (n ≥ 22) is plotted. Chi-square analysis was used to compare between groups. Experiments were performed at 20°C. Source data are provided in S1 Table.
(TIF)

**S7 Fig. Knock-down of genes important for sheath cell development in *daf-2*.** Representative images showing that *lim-7* and *lim-6* RNAi do not result in sterility in *daf-2(e1370)* worms. The upper panel shows brightfield images where yellow arrowheads mark eggs. In the DIC images of the lower panel, white arrows point towards the oocytes in the gonadal arm.
(TIF)

**S8 Fig. Effect of *cyd-1* knock-down on male gonad marker expression in *daf-2* hermaphrodites and males.** Representative fluorescent and DIC merged images of gonads showing *K09C8.2*::GFP (male gonad marker) expression in *daf-2(e1370)* hermaphrodite or male worms (day-1 adult) grown on control and *cyd-1* RNAi. Scale bar 20 μm.
(TIF)

**S1 Table. Source Data for all experiments used in the study.**
(XLSX)

**S2 Table. A list of differentially expressed genes in *daf-2* gonad-specific *cyd-1* KD.**
(XLSX)

**S3 Table. A list of differentially expressed genes in wild-type gonad-specific *cyd-1* KD.**
(XLSX)

**S4 Table. A list of quantitative Real-time PCR primers.**
(XLSX)

## Acknowledgments

We thank Dr. Anita Goyala, Ranjisha KR, Oviya Devendran and Nikhita Anand for their help with experiments, and all members of Molecular Aging Laboratory for their support. The SU93 strain *jcIs1[AJM-1::GFP + unc-29(+) + rol-6(su1006)] IV* was kindly provided by Dr. K. Subramaniam and we also thank him for his valuable suggestions which have furthered our understanding in this study. Some strains were provided by the CGC, which is funded by NIH Office of Research Infrastructure Programs (P40 OD010440).

## Author Contributions

**Conceptualization:** Umanshi Rautela, Gautam Chandra Sarkar, Arnab Mukhopadhyay.

**Data curation:** Umanshi Rautela, Gautam Chandra Sarkar, Mohtashim Rosh, Aneeshkumar G. Arimbasseri.

**Formal analysis:** Umanshi Rautela, Gautam Chandra Sarkar, Ayushi Chaudhary, Mohtashim Rosh, Aneeshkumar G. Arimbasseri.

**Funding acquisition:** Arnab Mukhopadhyay.

**Investigation:** Umanshi Rautela, Gautam Chandra Sarkar, Ayushi Chaudhary, Debalina Chatterjee, Mohtashim Rosh, Aneeshkumar G. Arimbasseri.

**Methodology:** Umanshi Rautela, Gautam Chandra Sarkar.

**Project administration:** Arnab Mukhopadhyay.

**Supervision:** Aneeshkumar G. Arimbasseri, Arnab Mukhopadhyay.

**Validation:** Umanshi Rautela, Gautam Chandra Sarkar.

**Visualization:** Umanshi Rautela, Gautam Chandra Sarkar, Ayushi Chaudhary, Aneeshkumar G. Arimbasseri.

**Writing – original draft:** Umanshi Rautela, Gautam Chandra Sarkar, Arnab Mukhopadhyay.

**Writing – review & editing:** Umanshi Rautela, Gautam Chandra Sarkar, Arnab Mukhopadhyay.

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
