## [Decision Letter · Decision Letter 0]

22 Aug 2024

Dear Dr Mukhopadhyay,

Thank you very much for submitting your Research Article entitled 'A non-canonical role of somatic Cyclin D/CYD-1 in oogenesis and reproductive aging, dependent on the FOXO/DAF-16 activation state' to PLOS Genetics.

The manuscript was fully evaluated at the editorial level and by independent peer reviewers. The reviewers appreciated the attention to an important topic but identified some concerns that we ask you address in a revised manuscript.

We therefore ask you to modify the manuscript according to the review recommendations. Your revisions should address the specific points made by each reviewer.

To resubmit, log into your Editorial Manager account and select the option 'Revise Submission' in the 'Submissions Needing Revision' folder.

Yours sincerely,

Fengwei Yu

Section Editor

PLOS Genetics

Fengwei Yu

Section Editor

PLOS Genetics

Reviewer's Responses to Questions

**Comments to the Authors:**

Reviewer #1: All of my prior comments have been carefully addressed. I find no outstanding issues to correct.

Reviewer #2: The authors made an effort to address most of my concerns. However, they reinforced my concern about their interpretation that the observed effects show reproductive senescence. The additional data provided confirms that the observed oocyte defects are rather due to developmental defects than premature aging. cyd-1 RNAi in L4 does not reduce oocyte quality. Animals exposed to cyd-1 RNAi in L1 exhibit lower overall egg production and do not stop egg laying premature. Furthermore, these animals show never a “young phenotype”- quite the contrary, all results point to developmental defects that lead to reproductive defects that get worse with age- but "sick" getting "sicker" is not a reproductive aging phenotype.

I also want to point out that data that is discussed in the paper needs to be included for the reader. So, I "insist" to include the data from former page 13 as wells page 22 (now page 23 "We inquired if knocking down genes essential for sheath cell development (e.g., 23 lim-7, lim-6) would also lead to germline arrest in daf-2 but found no such sterility (data not shown).") that was not shown in the first version of the manuscript.

Reviewer #3: I agree with the previous reviewer 1's comments that the additional data provided support a developmental role for cyd-1 in oogenesis, and not reproductive aging (at day 1 the fertility of cyd-1 RNAi is already lower than control). I suggest the authors to tone down their conclusions to avoid any suggestions on a reproductive aging phenotype.

**Have all data underlying the figures and results presented in the manuscript been provided?**

Reviewer #1: Yes

Reviewer #2: **No: **I asked in my first review to include data sets on page 13 and page 22 (now page 23 "We inquired if knocking down genes essential for sheath cell development (e.g.,

23 lim-7, lim-6) would also lead to germline arrest in daf-2 but found no such sterility (data not shown).") that are not shown.

They added the data from page 13 in the revised manuscript, but only showed the data from page 22 in their response to the reviewers. These need to be added to the manuscript.

Reviewer #3: Yes

PLOS authors have the option to publish the peer review history of their article (what does this mean?). If published, this will include your full peer review and any attached files.

Reviewer #1: No

Reviewer #2: No

Reviewer #3: No

---

## [Decision Letter · Decision Letter 1]

7 Oct 2024

Dear Dr Mukhopadhyay,

We are pleased to inform you that your manuscript entitled "A non-canonical role of somatic Cyclin D/CYD-1 in oogenesis and in maintenance of reproductive fidelity, dependent on the FOXO/DAF-16 activation state" has been editorially accepted for publication in PLOS Genetics. Congratulations!

Yours sincerely,

Fengwei Yu

Academic Editor

PLOS Genetics

Fengwei Yu

Section Editor

PLOS Genetics

Comments from the reviewers (if applicable):

All the reviewers' comments have been addressed.

Reviewer's Responses to Questions

**Comments to the Authors:**

Reviewer #2: My concerns are addressed.

Reviewer #3: The authors have addressed my comments.

**Have all data underlying the figures and results presented in the manuscript been provided?**

Reviewer #2: Yes

Reviewer #3: Yes

PLOS authors have the option to publish the peer review history of their article (what does this mean?). If published, this will include your full peer review and any attached files.

Reviewer #2: No

Reviewer #3: No

**Data Deposition**

http://datadryad.org/submit?journalID=pgenetics&manu=PGENETICS-D-24-00562R1

**Press Queries**

---

## [Editor Report · Acceptance letter]

8 Nov 2024

PGENETICS-D-24-00562R1 

A non-canonical role of somatic Cyclin D/CYD-1 in oogenesis and in maintenance of reproductive fidelity, dependent on the FOXO/DAF-16 activation state 

Dear Dr Mukhopadhyay, 

We are pleased to inform you that your manuscript entitled "A non-canonical role of somatic Cyclin D/CYD-1 in oogenesis and in maintenance of reproductive fidelity, dependent on the FOXO/DAF-16 activation state" has been formally accepted for publication in PLOS Genetics! Your manuscript is now with our production department and you will be notified of the publication date in due course.

With kind regards,

Anita Estes

PLOS Genetics

On behalf of:
